# LLaVA-Plus: Learning to Use Tools for Creating Multimodal Agents

## Abstract

This paper presents LLaVA-Plus (**L**arge **L**anguage **a**nd **V**ision **A**ssistants that **P**lug and **L**earn to **U**se **S**kills), a general-purpose multimodal assistant trained using an end-to-end approach that systematically expands the capabilities of large multimodal models (LMMs). LLaVA-Plus maintains a skill repository that contains a wide range of vision and vision-language pre-trained models (tools), and is able to activate relevant tools, given users' multimodal inputs, to compose their execution results on the fly to fulfill many real-world tasks. To acquire the ability of using tools, LLaVA-Plus is trained on multimodal instruction-following data that we have curated. The training data covers many tool use examples of visual understanding, generation, external knowledge retrieval and their compositions. Empirical results show that LLaVA-Plus outperforms LLaVA in existing capabilities, and exhibits many new capabilities. Compared with tool-augmented LLMs, LLaVA-Plus is distinct in that the image query is directly grounded in and actively engaged throughout the entire human-AI interaction sessions, significantly improving tool use performance and enabling new scenarios.

## 1 Introduction

A long-standing aspiration in artificial intelligence is to develop general-purpose assistants that can effectively follow users' (multimodal) instructions to complete a wide range of real-world tasks (Askell et al., 2021; Li et al., 2023c). Recently, the community has witnessed a growing interest in developing foundation models with emergent abilities of multimodal understanding and generation in open-world tasks (Gan et al., 2022; Li et al., 2022). While the recipes of using Large Language Models (LLMs) such as ChatGPT (OpenAI, 2023a) to develop general-purpose assistants for natural language tasks have been proved effective in many tasks, the recipes of building general-purpose, multimodal assistants for computer vision and vision-language tasks remain to be explored.

Ongoing efforts of developing multimodal agents can be broadly categorized into two classes (Li et al., 2023c): ($i$) *End-to-end training with LLMs*, where image-text data and multimodal instruction-following data are collected to continually train LLMs to acquire the ability of processing visual information, resulting in a series of Large Multimodal Models (LMMs). Impressive visual understanding and reasoning performances have been demonstrated by both proprietary models such as Flamingo (Alayrac et al., 2022) and multimodal GPT-4 (OpenAI, 2023c), and open-sourced models such as LLaVA (Liu et al., 2023a) and MiniGPT-4 (Zhu et al., 2023). Although these end-to-end training methods are effective in helping LMMs to gain emergent abilities (such as in-context learning), it remains challenging to develop a unified architecture that can seamlessly incorporate a wide range of skills, such as image segmentation and generation, which are crucial for real-world multimodal applications. ($ii$) *Tool[1] chaining with LLMs*, where the prompts are meticulously crafted to enable LLMs (*e.g.,* through LangChain lan (2022)) to invoke different tools (*e.g.,* pre-trained vision models) to perform desired (sub-)tasks, without the need of additional model training. Some prominent works include VisProg (Gupta & Kembhavi, 2022), ViperGPT (Surís et al., 2023), Visual ChatGPT (Wu et al., 2023), X-GPT (Zou et al., 2023a), and MM-REACT (Yang et al., 2023b). The strength of these methods is the ability to perform a broad spectrum of visual tasks through the use of (new) tools, which can be incorporated into an AI agent with very low development cost. However, prompting is neither adaptable nor robust enough to allow multimodal agents to always accurately select and

---

[1]The term "tools" in this paper is used to describe the APIs or pre-built models that LMM interfaces with.

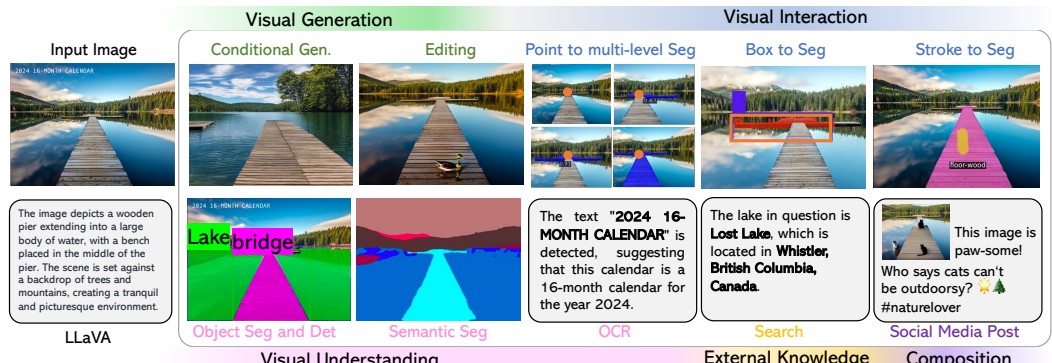

Figure 1: Visual illustration of LLaVA-Plus' capabilities enabled by learning to use skills.

activate appropriate tools (from a large and diverse toolset) and compose their results to generate final answers on the fly for real-world multimodal tasks.

In this paper, we present LLaVA-Plus (**L**arge **L**anguage **a**nd **V**ision **A**ssistants that **P**lug and **L**earn to **U**se **S**kills), a general-purpose multimodal assistant that learns to use tools using an end-to-end training approach that systematically expands the capabilities of LMMs via visual instruction tuning. To the best of our knowledge, this is the first attempt reported to combine the strengths of the end-to-end training and tool chaining methods mentioned above. LLaVA-Plus is equipped with a skill repository that contains a wide range of vision and vision-language tools. The design is an embodiment of the "Society of Mind" scheme (Minsky, 1988), where each tool is originally designed for a specific skill and by itself is only useful for specific scenarios, but the combinations of these tools lead to emergent abilities that show signs of higher intelligence. For example, LLaVA-Plus is able to construct a new workflow on the fly, given users' multimodal inputs, select and activate relevant tools from the skill repository, and compose their execution results to fulfill many real-world tasks that are unseen during model training.

LLaVA-Plus can be continually improved by incorporating new skills or tools via instruction tuning. Consider a new multimodal tool that has been developed for a specific scenario or skill. We collect pertinent user instructions that request this tool and their execution results (or following) to form instruction-following data for tuning. After instruction tuning, LLaVA-Plus expands its abilities as it learns to use this new tool to deal with the tasks that it cannot handle before. LLaVA-Plus also differs from those existing works on teaching LLMs to use tools (*e.g.,* Yang et al., 2023a; Patil et al., 2023), where visual signals are only used when the multimodal tools are activated. In contrast, LLaVA-Plus uses the raw visual signals through the entire human-AI interaction sessions to improve LMM's ability of planning (determining the most appropriate tools to use for a given task) and reasoning.

In summary, our paper makes the following contributions:

- *New multimodal instruction-following tool use data*. We present a new pipeline for curating vision-language instruction-following data, dedicated for tool use in human-AI interaction sessions, leveraging ChatGPT and GPT-4 as labeling tools.
- *New large multimodal assistant*. We have developed LLaVA-Plus, a general-purpose multimodal assistant that extends LLaVA (Liu et al., 2023a) by incorporating a large and diverse set of external tools that can be selected, composed, and activated on the fly for performing tasks. As shown in Figure 1, LLaVA-Plus significantly extends LMM's capabilities. Our empirical study validates the effectiveness of LLaVA-Plus with consistently improved results on multiple benchmarks, and in particular, new SoTA on VisiT-Bench with a diverse set of real-life tasks.
- *Open-source*. We will release the following assets to the public: the generated multimodal instruction data, the codebase, the LLaVA-Plus checkpoints, and a visual chat demo.

## 2 LEARNING TO USE TOOLS WITH VISUAL INSTRUCTION TUNING

### 2.1 PRELIMINARIES: VISUAL INSTRUCTION TUNING IN LLAVA

Inspired by the impressive performance of multimodal GPT-4 and the open-source LMMs such as LLaVA/MiniGPT-4, the community has witnessed a surge in developing LMMs and the multimodal

instruction-following data, following the instruction tuning paradigm (*e.g.,* Liu et al., 2023a; Peng et al., 2023a). In this paper, we use LLaVA as a running example. But note that the proposed recipe can be easily applied to other LMMs. Starting with a user input image query $\mathbf{I}_q$, existing LMMs such as LLaVA typically accept a natural language instruction input $\mathbf{X}_q$ from the user, and output a natural language response $\mathbf{X}_{\texttt{answer}}$. Therefore, we can use a unified scheme to represent multimodal instruction-following data as:

$$\texttt{Human}: \mathbf{I}_q <\backslash n> \mathbf{X}_q \texttt{<STOP> Assistant}: \mathbf{X}_{\texttt{answer}}\texttt{<STOP>}, \qquad (1)$$

where `Human` and `Assistant` are special role tokens, $<\backslash n>$ and `<STOP>` are the line break token and sequence end token, respectively. It naturally covers any multimodal tasks that can be formulated as language-image input and language output, ranging from simple visual understanding tasks such as recognition, captioning, and visual question answering (VQA) to complex visual reasoning tasks. Due to its simplicity, the data pipeline is easy to construct and scale. By training a single Transformer-based model with an auto-regressive objective, the resulting LMM enables a seamless human-assistant interaction, proficiently completing many visual tasks in the wild. However, it is limited in flexibility regarding skill expansion and engagement in human-AI interactions.

## 2.2 LLaVA-PLUS

We propose a modularized system architecture that allows an LMM, working as a planner, to learn to use a wide range of skills at scale, and thus facilitating easy expansion of its capabilities and interface. Specifically, we build a skill repository, where the LMM can leverage a broad range of existing vision and vision-language specialist models as tools for their respective skills when needed, to complete various tasks in the wild. The LMMs in most existing multimodal agents typically perform *user-oriented dialogues*, where the LMMs are required to immediately respond to user instructions based solely on the knowledge encoded in model weights, as shown in equation 1 and the left part of Figure 2. In addition to this, the LMM in LLaVA-Plus also performs *skill-oriented dialogues*, where the LMM initiates requests to call appropriate tools from the skill repository, and subsequently aggregate the tool execution results after applying proper skills, as shown in the right part of Figure 2.

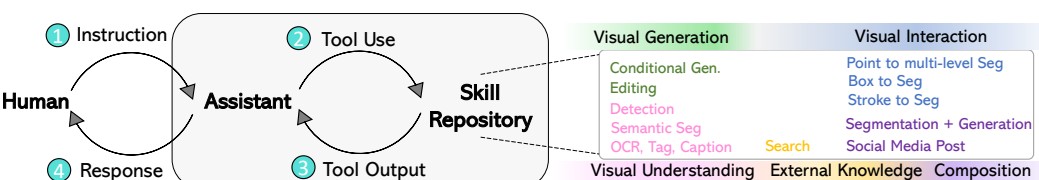

Figure 2: The four-step LLaVA-Plus pipeline. The skill repository is shown on right.

**A Full Dialogue of LLaVA-Plus.** We illustrate how LLaVA-Plus works with a full dialogue session in Figure 2. It proceeds in four steps: ① Humans provide a task instruction $\mathbf{X}_q$ related to an image $\mathbf{I}_q$. ②The LMM-powered assistant analyzes both $\mathbf{X}_q$ and $\mathbf{I}_q$, and outputs $\mathbf{X}_{\texttt{skill\_use}}$ that chooses the tool from skill repository and writes the appropriate prompt as the tool argument. ③ By executing the tool, the result $\mathbf{X}_{\texttt{skill\_result}}$ is returned to the assistant. ④ The assistant aggregates $\mathbf{X}_{\texttt{skill\_result}}$ with $\mathbf{X}_q$ and $\mathbf{I}_q$, and outputs $\mathbf{X}_{\texttt{anwser}}$ to humans. The interaction can be represented as:

$$\texttt{Human}: \mathbf{I}_q <\backslash n> \mathbf{X}_q \texttt{<STOP> Assistant}: \mathbf{X}_{\texttt{skill\_use}}\texttt{<STOP>}$$
$$\texttt{Human}: \mathbf{X}_{\texttt{skill\_result}}\texttt{<STOP> Assistant}: \mathbf{X}_{\texttt{anwser}}\texttt{<STOP>} \qquad (2)$$

Compared with equation 1 which is used to train LLaVA, the only newly introduced component for LLaVA-Plus training is the skill-oriented dialogue. Table 1 illustrates one sequence example of calling detection and segmentation skills in human-AI interactions. LLaVA-Plus is trained with an auto-regressive objective on the sequence of equation 2, where only the green sub-sequences (or tokens) are used to compute the loss, and thus the model learns to predict skill use, answers, and when to stop.

**Unified Prediction Format from LMMs.** Figure 2 shows that the LMM of LLaVA-Plus needs to perform both user-oriented and skill-oriented dialogues. To this end, we use a unified model prediction format to represent dialogues with and without the need of calling the skill repository. Inspired by Yao et al. (2022), the format consists of three fields, as illustrated in Table 1: (*i*) `Thought` is a text

| Human $(\mathbf{X}_q, \mathbf{I}_q)$ | Can you segment the girl and the cart in the image and describe their relationship? *(The source image from COCO with ID 569536)* |
|---|---|
| Assistant $(\mathbf{X}_{\mathtt{skill\_use}})$ | **"thoughts"** To accurately segment and identify the girl and the cart in the image, we can leverage the grounding_dino model for open-set detection and the segmentation model SAM. Integrating these models will enable us to generate a comprehensive summary of the segmented subjects. **"actions"** ["API_name": "grounding_dino+sam", "API_params": "caption": "little girl and cart ."] **"value"** I will use grounding_dino+sam to help to answer the question. Please wait for a moment. |
| *Skill Repo.* Human $(\mathbf{X}_{\mathtt{skill\_result}})$ | *Tool execution: Grounding DINO + SAM* grounding_dino+sam model outputs: 'boxes': [[0.35, 0.37, 0.66, 0.97], [0.0, 0.57, 0.69, 1.0]], 'logits': [0.58, 0.41], 'phrases': ['little girl', 'cart'] Can you segment the girl and the cart in the image and describe their relationship? |
| Assistant $(\mathbf{X}_{\mathtt{anwser}})$ | **"thoughts"** Thanks to the output of grounding_dino+sam. I can answer the question better. **"actions"** [] **"value"** Sure! I segment and identify the little girl and the cart in the image.
 - Centered in the image, a little girl is discernible with bounding box coordinates [0.35, 0.37, 0.66, 0.97]. She's seen actively maneuvering a cart within the airport's ambiance.
 - To the bottom left of the image, a cart can be pinpointed, bounded by the coordinates [0.0, 0.57, 0.69, 1.0]. It's laden with various luggage items, indicating travel preparations or arrivals.
 The interaction between the girl and the cart epitomizes a typical airport scenario: a traveler utilizing the cart to facilitate her movement with luggage. This tableau, whether signifying an arrival or departure, encapsulates the vibrancy and expectation inherent to air travel. |

Table 1: An example of a LLaVA-Plus workflow that plugs and learns to use the skills of object detection and segmentation, enhanced by a rich region language description. The gray text is not in the training sequence.

sequence representing a reasoning process, which determines whether the skill repository is needed to follow the user instruction, and if so, which tools to use. $(ii)$ `Action` is a list of function calls for the tools to execute the `thought`. The list is in the JSON format, with each item consisting of two sub-fields: `API_name` to call the tool and `API_params` for the corresponding function arguments if applicable. When `action` is an empty list, no skill is invoked. $(iii)$ `Value` is a natural language response that LLaVA-Plus generates by aggregating tool execution results and the human-AI session history. When presented in $\mathbf{X}_{\mathtt{skill\_use}}$ of user-oriented dialogues, it is the final response returned to human users. When presented in $\mathbf{X}_{\mathtt{anwser}}$ of skill-oriented dialogues, it is a natural language description about tool execution. In the serving stage, we find it important to ensure a good user experience that we only return the content in the `value` field of $\mathbf{X}_{\mathtt{anwser}}$ to human users, but hide the entire skill-oriented dialogues unless we need to debug the system.

## 2.3 Skill Repository: Multimodal Tool Use Instruct Data Generation

The skill repository of LLaVA-Plus consists of multimodal tools of different skills. To allow the LMM to always activate the most appropriate tools to complete a task, the corresponding tool-use multimodal instruction-following data is needed for LMM tuning. In alignment with the LLaVA approach, we input image information into a text-only GPT-4 model, prompting it to generate both questions and responses based on the visual data. Without loss of generality, in this study we want LLaVA-Plus to deal with the scenarios that requires novel skills that LLaVA does not have, *e.g.,* the individual skills for visual understanding, generation, and external knowledge retrieval and the compositions of these individual skills, as summarized in Table 2. In what follows, we treat visual understanding skills as core skills and the others as extended skills, and describe the way instruction data is curated.

### 2.3.1 Core Skills: Understanding

Visual understanding skills enable machines to interpret and comprehend visual signals. Existing LMMs have only a limited subset of visual understanding skills, constrained by language inputs and outputs. We expand them to a broader skill set with visual input prompts and visual outputs,

| | Skills | | Tools | Source | Size |
|---|---|---|---|---|---|
| *Individual Skills* | Understanding | Detection/Grounding | G-DINO (Liu et al., 2023b) | COCO | 13783 |
| | | Semantic Segmentation | OpenSeeD (Zhang et al., 2023a) | COCO | 5989 |
| | | Instance Segmentation | G-DINO+SAM | COCO | 5228 |
| | | Caption + Grounding | BLIP2+G-DINO | COCO | 4037 |
| | | Tagging + Grounding | RAM+G-DINO | COCO | 4439 |
| | | Caption | BLIP2 Li et al. (2023e) | COCO | 4064 |
| | | Tagging | RAM (Zhang et al., 2023d) | COCO | 6045 |
| | | OCR | EasyOCR (JaidedAI, 2023) | Hiertext | 6528 |
| | External Knowledge | Retrieval | CLIP Retrieval (Radford et al., 2021) | InfoSeek | 4087 |
| | Generation | Image Generation | Stable Diffusion (Rombach et al., 2021) | JourneyDB | 4694 |
| | | Image Editing | Instruct P2P (Brooks et al., 2023) | Instruct P2P | 6981 |
| | Visual Prompt | Interactive Segmentation | SAM (Kirillov et al., 2023) | COCO | 5601 |
| | | Multi-granularity | Semantic SAM (Li et al., 2023d) | COCO | 5601 |
| | | Example Based Segmentation | SEEM (Zou et al., 2023b) | COCO | 5601 |
| *Composed Skills* | Mix of Detection, Segmentation, Tagging, Caption | | G-DINO, SAM, BLIP2, RAM | COCO | 37,431 |
| | Interactive Segmentation + Inpainting | | SAM + Stable Diffusion | COCO | 3063 |
| | Semantic Segmentation + Generation | | OpenSeeD + ControlNet (Zhang et al., 2023b) | COCO | 5989 |
| | Image Generation + Social Media Post | | Stable Diffusion | JourneyDB | 4694 |
| | Image Editing + Social Media Post | | Instruct P2P Brooks et al. (2023) | Instruct P2P | 5924 |

Table 2: LLaVA-Plus skill repository and dataset statistics of our created visual instruction-following data for each tool use case. G-DINO indicates Grounding DINO (Liu et al., 2023b). HierText (Long et al., 2022; 2023), InfoSeek (Chen et al., 2023b), and JourneyDB (Pan et al., 2023) are datasets for OCR, external knowledge, and image generation, respectively.

including open-set detection and grounding, semantic/instance/interactive segmentation, tagging, captioning, OCR and their compositions, and so on. These understanding skills can be grouped into two categories, depending on whether additional function arguments are required.

**Skills with Image-only.** The skills without additional function arguments include captioning, tagging, semantic segmentation, caption+grounding, tagging+grounding, and OCR. We have curated training samples for each tool individually. To collect the training samples for a given skill, we fill in the four data variables in equation 2 using different strategies. $(i)$ For $\mathbf{X}_{\mathtt{q}}$, we use GPT-4 to generate a set of instructions that require the use of tools for proper answers. For each sample, we randomly select a question and rewrite it to enhance data diversity. An rewriting example is shown in Table 9 in Appendix. $(ii)$ For $\mathbf{X}_{\mathtt{skill\_use}}$, its $\mathtt{thoughts}$ and $\mathtt{value}$ are generated by randomly selecting from some preset responses with rewriting. The $\mathtt{actions}$ is known, so it can be directly assigned. $(iii)$ $\mathbf{X}_{\mathtt{skill\_result}}$ is generated with a fixed rule: first presenting the tool outputs and then repeating the initial question. $(iv)$ For $\mathbf{X}_{\mathtt{anwser}}$, its $\mathtt{thoughts}$ is created in a similar way to $\mathtt{thoughts}$ in $\mathbf{X}_{\mathtt{skill\_use}}$, and $\mathtt{action}$ is set empty. The $\mathtt{value}$ of $\mathbf{X}_{\mathtt{anwser}}$ is the most important field, as it is the visible response to humans in chat. We feed all previous information, including previous questions, the previous tool outputs, and context of the image to language-only GPT-4, which then generates responses to form instruction-following data. Inspired by LLaVA, we consider the ground-truth captions, object coordinates, and object categories as image contexts.

**Skills with Additional Function Arguments.** Visual skills such as object detection and instance segmentation often require humans to provide very specific instructions regarding the concepts of interests. Their instruction-following data is more challenging to create. We use two methods in this study. $(i)$ The first method is similar to that in the image-only skill setting, where the initial $\mathbf{X}_{\mathtt{q}}$ contains a placeholder $\mathtt{concept}$, one or more categories presented in the image are randomly chosen to replace this placeholder, and the final $\mathbf{X}_{\mathtt{q}}$ is obtained via rewriting, as shown in Table 9. $(ii)$ To allow the LMM to learn more diverse prompts beyond category information, we use GPT-4 to generate questions. Specifically, we manually create two seed samples following the full dialogue in equation 2, send them, together with image contexts, to GPT-4, and ask GPT-4 to generate a full dialogue based on a new image context. An example is shown in Table 10 in Appendix.

### 2.3.2 EXTENDED SKILLS

The LLaVA-Plus recipe can be applied to any tools to improve the system capabilities. We demonstrate its versatility by onboarding multimodal tools of different categorizes. Due to the limited space, we describe the instruction-following data creation process in Section B in Appendix, and summarize the extended skills we have enabled.

- **External Knowledge.** To enable LMMs to use knowledge beyond that encoded in pre-trained model weights, we use the CLIP search API to retrieve external knowledge from LIAON.
- **Generation.** To allow LLaVA-Plus to output images, we use Stable Diffusion (SD) and Instruct-Pix2Pix for image generation and editing, respectively.
- **Visual Prompts.** To better follow human intents, we support various visual prompts for human-AI interaction, such as user-drawn points, sketches and boxes. SAM, Semantic-SAM and SEEM are used for different interactive segmentation tasks.
- **Skill Composition.** To allow LLaVA-Plus to deal with real-world compositional tasks. We curate data for the following scenarios: $(i)$ The scenarios where various visual understanding results of the same image in a multi-turn human-AI interaction session are required. We generate instruction data by applying different tools (including detection, segmentation, tagging, and captioning). $(ii)$ Interactive Segmentation + Inpainting. By combining the SAM segmentation results from the user pointing and SD, we enable inpainting with visual interaction. $(iii)$ Semantic Segmentation + Generation. By combining the spatial layout from OpenSeed semantic segmentation and ControlNet, we enable instructional visual-conditioned generation. $(iv)$ Image Generation/Editing + Social Media Post. It is time-consuming for human users to generate posts that contains both images and text. Thus, we use SD to generate an image, or Instruct Pix2Pix to edit an image, then combine the image with its description generated by a pre-trained LMM to create a multimodal post.

## 2.4 MODEL TRAINING AND SERVING

**Training.** To train LLaVA-Plus, we combine the curated tool use instruction data, as shownin Table 2, with the LLaVA-158K dataset. To convert LLaVA-158K into the unified prediction format as described in Section 2.2, we treat the responses in LLaVA-158K as `value`, and add the fields of `thoughts` and `actions` with templates, as illustrated in the example in Table 8 in Appendix. LLaVA-Plus are built in two settings. $(i)$ *LLaVA-Plus (All Tools), where tool use is cast as external knowledge*. All visual understanding tools except segmentation in Table 2 are utilized to process the input image, and the extracted recognition results are organized as symbolic sequence representations to enrich the image features in both the training and evaluation stages. $(ii)$ *LLaVA-Plus (Fly), where tools are used on the fly*. To reduce the cost of calling all tools, we only provide the execution results of related tools for a given instruction. When reporting quantitative numbers, we train models on the 81K understanding instruction data, because existing benchmarks focus mainly on understanding capabilities. When building demo systems, we train our models on the full dataset.

**Serving.** LLaVA-Plus is served using the FastChat (Vicuna, 2023) system, which is composed of web servers that interface with humans, model workers that host the LMM and multiple tools, and a controller to coordinate the web-server and model workers. The 7B LLaVA-Plus and all the tools can be loaded and served in a 80G GPU.

## 3 RELATED WORKS

We summarize the connections and differences between LLaVA-Plus and existing general-purpose multimodal systems in Table 3, where only representative methods are shown due to space constraint. They can be broadly categorized into two classes as discussed below.

| Capabilities | Image Understanding | | | Knowledge | Image Gen. | Visual Interaction | Combined | Too Use | |
|---|---|---|---|---|---|---|---|---|---|
| Input | (Text, Image) | | | | | (Point, Box) | All | Allocator | Training |
| Output | Text | Box | Mask | Text | Image | (Text, Image, Mask) | All | | |
| MM-REACT | ✓ | | ✓ | | ✓ | | | LLM | |
| GPT4Tools | ✓ | ✓ | ✓ | | ✓ | | | LLM | ✓ |
| LLaVA-Plus | ✓ | ✓ | ✓ | ✓ | ✓ | ✓ | ✓ | LMM | ✓ |
| LLaVA/GPT-V | ✓ | | | | | | | | |
| Kosmos-2 | ✓ | ✓ | | | | | | | |
| CM3Leon | ✓ | | ✓ | ✓ | ✓ | | | | |

Table 3: Comparison with existing multimodal systems. The empty cells indicate inapplicable. "Allocator" indicates which base model is used to invoke the tools, and "Training" indicates whether model training is needed to enable tool use.

**AI Agents with Multimodal Tool Use.** There is a growing interest in exploring a paradigm of building general-purpose AI agents that synergistically leverage multiple tools with LLMs to solve sophisticated, open-world problems. The idea is originated in NLP to invoke general tools whose skills are lacked from LLM (*e.g.,* ToolFormer (Schick et al., 2023), ChatGPT-Plugin (OpenAI, 2023b)), and is recently extended to the multimodal space. There are two ways to leverage multimodal tools with the LLM as a planner to determine which tools to invoke: ($i$) tool chaining by prompt engineering and in-context-learning, such as Visual ChatGPT (Wu et al., 2023), MM-ReAct (Yang et al., 2023b), and ($ii$) instruction tuning of LLM with a focus on multimodal tool use, such as GPT4Tools (Yang et al., 2023a) and Gorilla (Patil et al., 2023). LLaVA-Plus represents the first work of utilizing the LMM as the planner for tool use, where image inputs are considered throughout the entire interaction sessions for improved user experience.

**Unified Multimodal Models with Versatile Capabilities.** Inspired by the success of a unified architecture of LLMs to complete many language tasks, the AI community has witnessed an increasing interest in building unified models with versatile multimodal capabilities. Proprietary models such as Flamingo (Alayrac et al., 2022) and multimodal GPT-4 (OpenAI, 2023c) (or GPT-4V (OpenAI, 2023d)) have demonstrated strong multimodal performance on zero-shot task transfer, which quickly inspired their open-source counterparts: LLaVA, MiniGPT-4, Open-Flamingo (Awadalla et al., 2023), Otter (Li et al., 2023a), to name a few. These LMMs can deal with the tasks with image-text input and text output. The capabilities have been extended to support the tasks with image-text output, such as image editing and segmentation, as demonstrated in CM3Leon (Yu & et al, 2023), Emu (Sun et al., 2023), and GILL (Koh et al., 2023). Bounding box outputs for grounding are recently supported, as shown in Kosmos-2 (Peng et al., 2023b), Shikra (Chen et al., 2023a) and DetGPT (Pi et al., 2023). GPT4ROI (Zhang et al., 2023c) allows users to select regions of interest with bounding boxes for human-AI visual chat. BubaGPT (Zhao et al., 2023) and LISA (Lai et al., 2023) use an extra referring segmentation model to enable the mask prediction capability. Compared with them, LLaVA-Plus enables a much wider range of multimodal skills and their compositions, as illustrated in Table 3.

## 4 EXPERIMENTS

### 4.1 THE EFFECTIVENESS OF LEARNING TO USE SKILLS

**Tool Use Improves Existing Capabilities.** We consider two benchmarks. LLaVA-Bench (Liu et al., 2023a) evaluates the visual chat of LMMs, with three types of questions: conversation, detailed description and visual reasoning. It consists of two datasets: the *COCO* set containing 30 COCO images and 90 chat questions, and the *In-the-Wild* set containing 24 web images with 60 questions. Language GPT-4 (gpt4-0314) is used to score the generated answers. The relative scores between the model output and gold response are reported. SEED-Bench (Li et al., 2023b) evaluates the image-level and instance-level perception and reasoning of LMMs, with 19K multi-choice questions. The results are shown in Table 4. Both LLaVA-Plus variants outperform LLaVA on these two benchmarks, demonstrating the effectiveness of adding visual recognition results of applying new skills in the LMM pipeline. LLaVA-Plus (All Tools) shows superior performance to LLaVA-Plus (Fly) because the former leverages more tools as additional contexts. We further conducted several ablations: ($i$) We tried to directly add the skill execution results in the testing stage of LLaVA, shown as the row of LLaVA (Tools in Test). The degraded performance compared with LLaVA demonstrates the necessity of learning to use skills in training. ($ii$) We removed `thoughts` in the unified data format and observed a performance drop, indicating chain-of-thoughts style data format is beneficial. ($iii$) GPT4Tools trains an LLM for multimodal tool use. Its lower performance indicates that visual instruction tuning of tool use in LLaVA-Plus is important.

**LLaVA-Bench (Tools).** To study the novel capabilities enabled by learning to use skills, we create an evaluation set LLavA-Bench (Tools), which measures four capabilities (grounding, tagging, caption, and OCR) with 10, 12, 12, and 10 samples in each. In Table 5, we also compare against the commercial visual chat systems such as Microsoft BingChat and Google Bard. LLaVA-Plus significantly outperforms the others on this benchmark, mainly because the other systems are not equipped with some of these capabilities. By comparing with chaining tools with GPT-4 (row of "All tools + GPT4") and MM-REACT, we demonstrate the advantage of training an open-source LMM as a planner for tool use.

| | LLaVA-Bench (COCO) | | | | LLaVA-Bench (In-the-Wild) | | | |
|---|---|---|---|---|---|---|---|---|
| | Conv. | Detail | Reasoning | All | Conv. | Detail | Reasoning | All |
| LLaVA | 82.0 | 69.1 | 92.6 | 81.2 | 42.6 | 51.9 | 68.9 | 57.1 |
| LLaVA (Tools in Test) | 56.2 | 67.9 | 53.3 | 59.1 | 40.7 | 48.1 | 51.2 | 47.5 |
| LLaVA-Plus (All Tools) | 81.6 | 74.5 | 95.7 | **83.9** | 65.5 | 56.8 | 79.1 | **69.5** |
| LLaVA-Plus (Fly) | 76.2 | 72.2 | 92.3 | 80.4 | 45.2 | 50.4 | 72.6 | 59.1 |
| LLaVA-Plus (Fly) (no `thoughts`) | 76.6 | 70.4 | 90.7 | 79.4 | 38.8 | 39.8 | 59.8 | 48.7 |
| GPT4Tools | 75.3 | 53.8 | 86.9 | 72.1 | 31.1 | 27.1 | 54.1 | 40.7 |

(a) LLaVA-Bench.

| | Scene | Identity | Attribute | Location | Counting | Spatial | Interact. | Reason. | Text | Average |
|---|---|---|---|---|---|---|---|---|---|---|
| LLaVA | 59.50 | 54.29 | 56.06 | 42.54 | 39.35 | 33.03 | 43.30 | 41.39 | 30.59 | 44.45 |
| LLaVA (Tools in Test) | 67.13 | 56.85 | 45.24 | 47.24 | 45.69 | 40.18 | 60.82 | 70.09 | 30.59 | 51.54 |
| LLaVA-Plus (All Tools) | 68.94 | 56.80 | 58.89 | 47.34 | 48.14 | 45.21 | 60.82 | 71.30 | 37.65 | 55.01 |
| LLaVA-Plus (Fly) | 68.43 | 56.47 | 59.69 | 45.40 | 41.68 | 44.14 | 59.79 | 69.49 | 34.12 | 53.25 |

(b) SEED-Bench.

Table 4: LLaVA-Plus variants improves LLaVA on two LMM benchmarks.

| | Grounding | Tagging | Caption | OCR | All |
|---|---|---|---|---|---|
| LLaVA | 47.1 | 87.1 | 77.0 | 23.6 | 58.7 |
| LLaVA (Tools in Test) | 41.7 | 48.5 | 72.0 | 31.9 | 48.5 |
| LLaVA-Plus (All Tools) | 89.3 | 94.4 | 96.7 | 48.8 | **82.3** |
| LLaVA-Plus (Fly) | 88.6 | 88.9 | 90.2 | 38.4 | 76.5 |
| Bard (0730) | 36.5 | 105.3 | 103.3 | 60.0 | 76.3 |
| Bing Chat (0730) | 56.0 | 84.0 | 96.0 | 44.8 | 70.2 |
| MM-REACT | 30.2 | 94.7 | 103.8 | 77.3 | 76.5 |
| All Tools + GPT4 | 77.5 | 95.6 | 95.2 | 39.3 | 76.9 |

Table 5: LLaVA-Bench (Tool Use).

## 4.2 COMPARISONS WITH SoTA LMM SYSTEMS

**MMVet** (Yu et al., 2023) contains 200 images and 218 questions, aiming to evaluate six core vision-language (VL) capabilities and their combinations. For evaluation, an LLM-based evaluator (`gpt4-0613`) is used to score open-ended outputs of different forms. The results are reported in Table 6. LLaVA-Plus consistently outperforms LLaVA on both 7B and 13B model sizes. The categories with most significant improvements are OCR and spatial, indicating the positive impact of the corresponding visual skills on LMM outputs.

| Model | Rec | OCR | Knowledge | Generation | Spatial | Math | Total |
|---|---|---|---|---|---|---|---|
| *Results of various open-source LMM on reported in the MM-VET paper (Yu et al., 2023)* | | | | | | | |
| OpenFlamingo-9B (Awadalla et al., 2023) | 24.6 | 14.4 | 13.0 | 12.3 | 18.0 | 15.0 | 21.8±0.1 |
| BLIP-2-12B (Li et al., 2023e) | 27.5 | 11.1 | 11.8 | 7.0 | 16.2 | 5.8 | 22.4±0.2 |
| LLaVA-7B (Liu et al., 2023a) | 28.0 | 17.1 | 16.3 | 18.9 | 21.2 | 11.5 | 23.8±0.6 |
| MiniGPT-4-14B (Zhu et al., 2023) | 29.9 | 16.1 | 20.4 | 22.1 | 22.2 | 3.8 | 24.4±0.4 |
| Otter-9B (Li et al., 2023a) | 28.4 | 16.4 | 19.4 | 20.7 | 19.3 | 15.0 | 24.6±0.2 |
| InstructBLIP-14B (Dai et al., 2023) | 30.8 | 16.0 | 9.8 | 9.0 | 21.1 | 10.5 | 25.6±0.3 |
| MM-ReAct-GPT-3.5 (Yang et al., 2023b) | 24.2 | 31.5 | 21.5 | 20.7 | 32.3 | 26.2 | 27.9±0.1 |
| LLaMA-Adapter v2-7B (Gao et al., 2023) | 32.9 | 20.1 | 19.0 | 20.1 | 22.9 | 3.9 | 31.4±0.1 |
| LLaVA-13B (V1.3, 336px) (Liu et al., 2023a) | 38.1 | 22.3 | 25.2 | 25.8 | 31.3 | 11.2 | 32.5±0.1 |
| MM-ReAct-GPT-4 (Yang et al., 2023b) | 33.1 | 65.7 | 29.0 | 35.0 | 56.8 | 69.2 | 44.6±0.2 |
| *Results with our own experiment runs* | | | | | | | |
| LLaVA-7B | 30.4 | 13.3 | 19.2 | 20.1 | 18.7 | 8.1 | 24.1±0.0 |
| LLaVA-Plus-7B (All Tools) | 30.5 | 23.6 | 20.5 | 22.5 | 28.5 | 7.7 | 27.5±0.3 |
| LLaVA-Plus-13B (All Tools, V1.3, 336px) | 37.5 | 29.4 | 22.3 | 24.5 | 37.3 | 11.5 | **35.0±0.0** |

Table 6: Performance of various open-source LMM on MM-VET. Note that MM-ReAct is not a single multimodal model, it is a system built on chaining visual tools via GPT-3.5 or GPT-4, which we append as a reference. Our experiment running on LLaVA-7B yields very similar scores with the same checkpoint reported in MM-VET paper, indicating that our evaluation pipelines are consistent.

**VisIT-Bench** (Bitton et al., 2023) is a real-world use oriented LMM benchmark, comprising 592 questions and 1,159 public images categorized into 70 instruction families. The results are shown in Table 7, which summarizes the battles between LMMs with GPT-analog human judgment. Elo ratings are computed by treating each pairwise human judgment as a "match". The difference between the

| Model | Model Size | ELO | Matches | Win(#Ratings) |
|---|---|---|---|---|
| Human Reference | | 1382 | 5880 | — |
| LLaVA-Plus | 13B | **1203** | 678 | **35.07**% (134) |
| LLaVA | 13B | 1095 | 5420 | 18.53% (475) |
| mPLUG-Owl | 7B | 1087 | 5440 | 15.83% (480) |
| LlamaAdapter-v2 | 13B | 1066 | 5469 | 14.14% (488) |
| Lynx | 8B | 1037 | 787 | 11.43% (140) |
| Idefics | 9B | 1020 | 794 | 9.72% (144) |
| InstructBLIP | 13B | 1000 | 5469 | 14.12% (503) |
| Otter | 8B | 962 | 5443 | 7.01% (499) |
| Visual GPT | | 941 | 5437 | 1.57% (510) |
| MiniGPT-4 | 11B | 926 | 5448 | 3.36% (506) |
| Octopus V2 | | 925 | 790 | 8.90% (146) |
| OpenFlamingo V1 | | 851 | 5479 | 2.95% (509) |
| PandaGPT | 13B | 775 | 5465 | 2.70% (519) |
| MultimodalGPT | | 731 | 5471 | 0.19% (527) |

Table 7: Current ELO rankings on ViSiT-Bench leaderboard as of Sept. 27th, 2023. Some model sizes are skipped due to unrevealed.

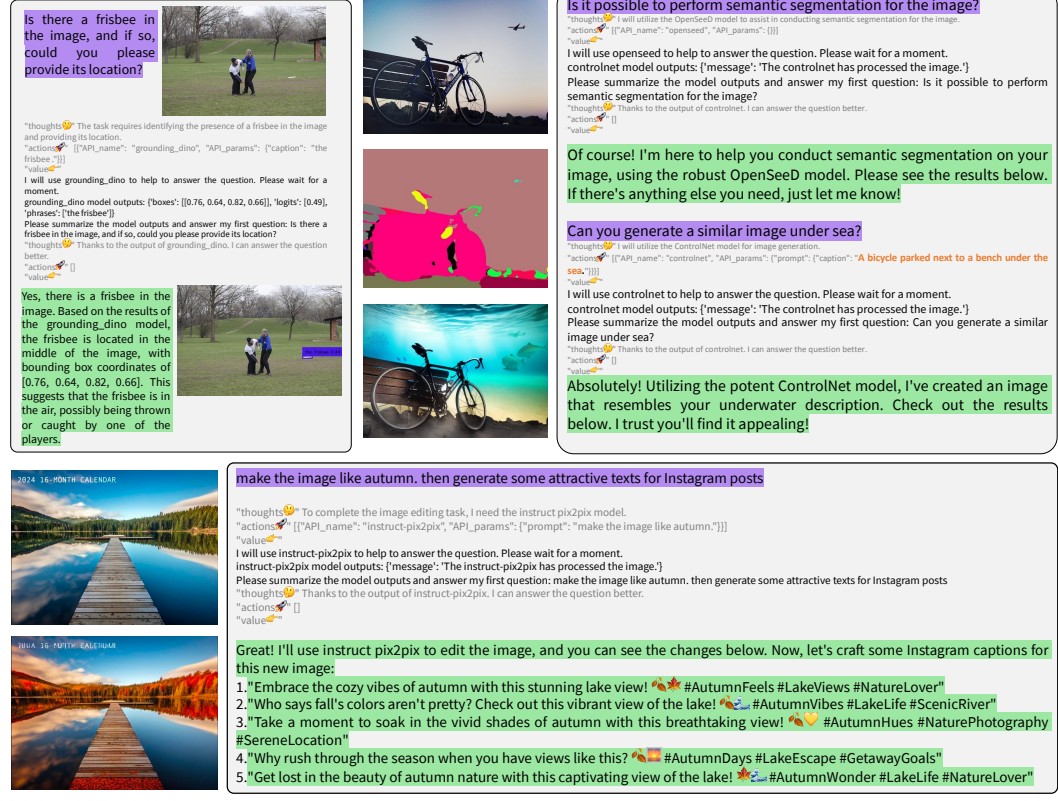

Figure 3: New capabilities in LLaVA-Plus. Human questions $\mathbf{X}_q$ are in purple, LLaVA-Plus responses $\mathbf{X}_{\texttt{anwser}}$ are in green. (Left) Object detection and visual chat; (Right) Semantic segmentation and mask-based conditional image generation; (Bottom) Multimodal social media post by editing an image and writing a message.

Elo ratings of two models provides an estimate for the win probability when pitting model A vs. model B. The "#matches" column indicates the number of total matches in which a particular model participates. Win-rate indicates the win rate of a model against the human-verified reference outputs. LLaVA-Plus significantly outperforms the leading method LLaVA by 100+ ELO score, achieving a new SoTA on the leaderboard.

### 4.3 VISUAL EXAMPLES OF NEW CAPABILITIES

In Table 3, we illustrate new capabilities of LLaVA-Plus with visual examples. Please see Section D in Appendix for many other interesting scenarios that demonstrate the versatile capabilities of LLaVA-Plus by learning to use skills and their compositions.

In the left example, the questions require identifying the precise object locations. LLaVA-Plus can successfully detect the frisbee's coordinates, which help determine its status of flying in the air and thus describe the outdoor scene/activity. The same example is shown to Bard, Bing Chat, MM-REACT and LLaVA in Figure 6 in Appendix. They all fail, revealing the lack of grounding ability.

In the right example, we illustrate an interactive image editing scenario, where users aim to see the spatial layout of the scene first and then generate an image of a similar layout, but with a new "under water" scene. The LMM not only applies the correct skills, but also generates a function argument "A bicycle parked next to a bench under the sea" for conditional image generation. This reveals the appealing property of LMM as a planner, as it can see the raw image, and provide necessary image analysis results throughout the human-AI interaction process. More such examples are in Appendix Figure 11.

In the bottom example, we show that LLaVA-Plus can be used to help create multimodal social media posts. For example, when capturing an image, the user wants to post the same image in an autumn scene and associate the image with some attractive text to post Instagram. LLaVA-Plus can use the editing skills to revise the image, and combine the context of visual images and their related language topics to suggest several caption options. In Appendix Figure 12, we create all four seasons for the same scenarios, and observe that LLaVA-Plus can follow the instruction to easily switch among them while consistently maintaining the original image cue.

## 5 CONCLUSION

We have presented LLaVA-Plus, a general-purpose, multimodal assistant which is based on an LMM that plugs and learns to use skills to complete a wide range of vision-language tasks in the wild. The first visual instruction dataset specifically designed for multimodal tool use has been collected for model training. By incorporating the execution results of new skills, LLaVA-Plus consistently outperforms LLaVA across many benchmarks, creates a new SoTA and shows emergent multimodal interaction capabilities. However, LLaVA-Plus is limited due to hallucinations and tool use conflicts in practice. There are interesting problems yet to be addressed in future research on building reliable general-purpose multimodal AI agents.

**Reproducibility** To ensure the reproducibility of our research, we will publicly release a comprehensive set of assets including the generated multimodal instruction data, our codebase, the LLaVA-Plus checkpoints, and a visual chat demo. Additionally, we have ensured complete transparency by elaborating on every facet of our training data collection and model training within this paper, as shown in Sec. 2.

**Limitations** Although our model has successfully extended the LLaVA abilities with tool use, we recognize several areas for improvement. Firstly, the accuracy of tool use, though improved, still requires refinement for more precise and reliable interactions. Secondly, our generated data, despite rigorous validation, contains instances of hallucinations, highlighting a need for further enhancements in data quality and model reliability. Lastly, the base language model used, while effective, may not represent the ultimate potential of our approach. A more advanced base model could potentially lead to even better results, suggesting an exciting direction for future research. These limitations provide a roadmap for ongoing development, underscoring our commitment to evolving and refining our model.

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

## A  DATA

**Augmenting LLaVA data.** The original LLaVA data only consists of questions and answers. We need to augment this data to make it match with our regular data format. We transformed the original answers in LLaVA into a part of the `values` field, then added an empty list for `actions`, and generated a `thoughts` using ChatGPT. The `thoughts` should indicate that the model can answer the question without invoking any tools. An example is shown in Table 8 in Appendix. We found the model cannot invoke tools if we did not unify the two data formats.

| | LLaVA Instruction Data | Augmented LLaVA Instruction Data |
|---|---|---|
| User LLaVA-Plus | What are the colors of the bus in the image? The bus in the image is white and red. | What are the colors of the bus in the image? **"thoughts"** The questions can be answered by the information in the context, without need any external tools. **"actions"** [] **"value"** The bus in the image is white and red. |

Table 8: Augmenting LLaVA instruction data.

**Details on data generation.** The pipeline to generate questions for visual prompts is shown in Table 4. The pipeline to generate questions with image-related parameters is shown in Table 5. An example of rewriting questions using GPT4 is shown in Table 9. The self-instruct example to generate multi-turn conversation for detection is shown in Table 10.

| **Prompts for ChatGPT/GPT4 to rewrite questions:** | | |
|---|---|---|
| Prompt | System | You will be given a question. Please rephrase the question to convey the same meaning but with greater diversity. The question is asking a model to generate a caption for an image. |
| | Human | generate a caption for the image |
| | AI | Please create a short phrase that captures the essence of this photo. |
| | Human | What would be an appropriate caption for this picture? |
| | AI | What caption would you write for this image? |
| | Human | Please examine this picture closely and provide a comprehensive caption that elucidates all of the pertinent subjects, actions, contexts, and nuances that are crucial for understanding what is being depicted. Be thorough and exact in your detailed textual description. |
| GPT | AI | Please study this photo carefully and generate a wordy caption that comprehensively describes what is going on, ensuring to include all pertinent subjects, actions, contexts, nuances, and other specifics necessary for comprehensively representing the essence of the image. |

Table 9: Examples of rewriting questions using GPT4.

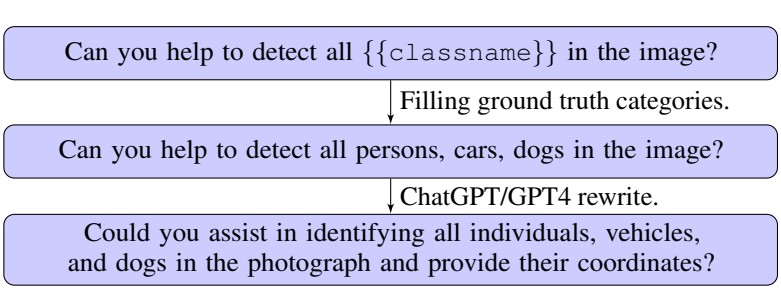

Figure 4: The pipeline to generate questions for visual prompts.

Figure 5: The pipeline to generate questions with image-related parameters.

| **Prompts for ChatGPT/GPT4 to rewrite questions:** | | |
| --- | --- | --- |
| Prompt | System | You are an AI visual assistant that can analyze a single image. You receive five sentences, each describing the same image you are observing. In addition, specific object locations within the image are given, along with detailed coordinates. These coordinates are in the form of bounding boxes, represented as (x1, y1, x2, y2) with floating numbers ranging from 0 to 1. These values correspond to the top left x, top left y, bottom right x, and bottom right y.
Design a conversation between you and a person asking about this photo. The answers should be in a tone that a visual AI assistant is seeing the image and answering the question.
In the first question question the user ask you to detect some objects in an image. To finish this, you will use an open-world object detection model grounding_dino to detect objects in the image.
The detection model used is grounding_dino. It is a model that can detect objects with captions in an image.
For each question, write an answer that you think would be appropriate. It must include the <thoughts>, the <actions>, and <actions>. <thoughts> is you thinking about the question and the answer. <actions> is what you do to answer the question. The API name should be "grounding_dino" or None.<values> is the value of the answer.
You must also include the output of the model, and how you would use it to answer the question.
Ask diverse questions and give corresponding answers. Only include questions that have definite answers. Do not ask any questions that cannot be answered confidently.
Give the answers as you have seen the image. Do not reveal the input information of the image. DO NOT say that you are given the captions and the objects in the image, JUST answer the question as if you are seeing the image for the first time, as well as the output of the model you invoked. |
| | Human | A plate filled with food sitting next to three glasses.
A plate of meats and cheese's next to three wine glasses.
A table displaying wine glasses and a platter of assorted meats and cheeses.
A white plate containing sliced ham and cheese sitting on a table next to wine glasses.
The wine has been poured and ready to drink
dining table: [0.0, 0.334, 0.893, 0.898]
person: [0.0003, 0.002, 0.111, 0.138]
person: [0.020, 0.0, 0.106, 0.109]
wine glass: [0.360, 0.109, 0.212, 0.501]
wine glass: [0.010, 0.107, 0.259, 0.612]
wine glass: [0.646, 0.101, 0.220, 0.487]
fork: [0.279, 0.588, 0.488, 0.225] |
| | AI | Question:
Is it feasible to call on a detection model to scan the image and generate a comprehensive summary of glasses?
Answer:
<thoughts> It would be beneficial to utilize an open-set detection model to identify the presence of glasses. Integrating the models into the conversation would enhance the overall outcome and provide more accurate results.
<actions> [{'API_name': 'grounding_dino', 'API_params': {'image': '<imagepath>', 'caption': 'glasses .'}}]
<values> Yes, it is feasible to leverage a detection model to scan the image and generate a comprehensive summary of glasses. In this case, I can utilize an open-set detection model called grounding_dino to detect all glasses present in the image.
Based on the bounding box coordinates detected, the image contains three wine glasses. Here's a summary of their positions:

1. The first wine glass is located towards the middle of the image, with bounding box coordinates of [0.360, 0.109, 0.212, 0.501]. This suggests that the glass is somewhat centrally placed on the table.
2. The second wine glass is located towards the left of the image, with bounding box coordinates of [0.010, 0.107, 0.259, 0.612]. This indicates that the glass is closer to the edge of the table or possibly near a person, given the presence of person bounding boxes in the image.
3. The third wine glass is located towards the right of the image, with bounding box coordinates of [0.646, 0.101, 0.220, 0.487]. This suggests that the glass is also closer to the edge of the table or near another person.

The bounding box coordinates suggest that all three glasses are roughly in a line, possibly indicating a formal dining setup. The glasses are also likely filled with wine.
======
Question:
Can a detection model be used to identify and summarize the presence of forks in the image?
Answer:
<thoughts> It would be beneficial to utilize an open-set detection model to identify the presence of forks. Integrating the models into the conversation would enhance the overall outcome and provide more accurate results.
<actions> [{'API_name': 'grounding_dino', 'API_params': {'image': '<imagepath>', 'caption': 'forks .'}}]
<values> Yes, it is feasible to leverage a detection model to scan the image and generate a comprehensive summary of forks. In this case, I can utilize an open-set detection model called grounding_dino to detect all forks present in the image.
Based on the bounding box coordinates detected, the image contains one fork. Here's a summary of its position:
1. The fork is located towards the middle of the image, with bounding box coordinates of [0.279, 0.588, 0.488, 0.225]. This suggests that the fork is placed near the plate of food, possibly indicating that someone is about to eat or is in the process of eating. |
| | Human | A girl with some food and drink at a table.
A YOUNG GIRL EATING PIZZA DRESSED AS A EGYPTION
A girl sitting around a table in a restaurant eating a pizza.
A young girl eating pizza in a restaurant.
A girl in a Cleopatra outfit eating pizza at a table.

chair: [0.394, 0.491, 0.248, 0.148]
dining table: [0.007, 0.573, 0.974, 0.416]
dining table: [0.663, 0.219, 0.101, 0.059]
person: [0.497, 0.229, 0.488, 0.634] |
| | AI | Question:
Perform an exhaustive search for the book in the image, and count the number of girls.
Answer:
<thoughts> Based on the information provided, the task requires me to search for the book in the image and count the number of girls.
<actions> [{'API_name': 'grounding_dino', 'API_params': {'image': '<imagepath>', 'caption': 'the book and the girls in the image.'}}]
<values> Certainly! To accomplish this task, I will utilize an object detection model to identify the book in the image and count the number of girls.
Based on the results of grounding_dino, I can identify the book and girls in the image. According to the available data, there is one 'book' object detected and there are three 'person' objects detected. Hence, there are three girls in the image.
======
Question:
Hi, can you tell me what's happening in this image?
Answer:
<thoughts> I don't need to utilize any specific tools to provide an answer to your question. I can answer the question by reading the image.
<actions> []
<value> Hello! According to the information provided, it seems like the image depicts a young girl, dressed as Cleopatra, who is eating pizza in a restaurant. |

Table 10: Self-instruct examples of generating multi-turns conversations with in-context learning.

**Data Filter Rules**   We recognize the potential for hallucination in datasets generated by models like GPT, and use some rules to clean the data:

1. *Removal of Apologetic Responses*: We have filtered out responses containing phrases like "I'm sorry", which often indicate uncertainty or errors in model-generated content.

2. *Exclusion of Language Model Content*: Entries containing references to "AI language model" or "language AI model" have been removed since we plan to train a multimodal model rather than a pure language model.

3. *Validation of CLIP-Retrieval Data*: For data pertaining to CLIP-Retrieval, we ensured that only items containing correct answers, as verified against ground-truth labels, were retained. Our selection criteria involve keeping items that match any of the listed ground truths.

4. *Accuracy in Object Detection and Segmentation Data*: We have excluded data where the provided answers contain bounding boxes that do not align with the established ground truths, ensuring accuracy in object detection and segmentation tasks.

## B   EXTENDED SKILLS

**External Knowledge.**   To enable LMMs to gain knowledge beyond that encoded in pre-trained model weights, we use the CLIP search API to retrieve external knowledge from LIAON. We utilize the images $I_q$ and questions $X_q$ from the InfoSeek dataset, and generate the other fields of the training sequence by following the image-only skill data creation pipeline. Input images are considered as queries, and image-to-text retrieval is performed to get top-K items for each query. To encourage the LMM to leverage external knowledge, we only consider the subset of questions whose ground truth answers can be extracted or derived from the retrieved knowledge. This subset can be selected using ChatGPT that compares the answers and retrieved knowledge.

**Generation.**   For image generation, we employ Stable Diffusion (SD) as the tool, and generate instruction data based on the JourneyDB dataset due to its high quality in language prompt and images. We ask ChatGPT to generate human-like instructions based on the original, detailed prompt for image generation, focusing on the scenarios where human-specified instructions are ambiguous and short, and thus cannot easily align with the prompt distribution of SD. Similarly, we use Instruct-Pix2Pix for image editing. The Instruct Pix2Pix dataset contains both instructions and prompts of source and target images. We directly use their editing instructions and follow the image-only skill data creation pipeline to fill the other fields.

**Visual Prompts.**   The visual prompt data is constructed similarly to that for visual understanding skills, except that additional visual inputs, such as user-drawn points, sketches and boxes, are required. Take SAM as an example. A point is required as input for interactive segmentation. We simply generate a random point and then convert it into a text sequence, and append it to a user question to form a concatenated text sequence $X_q$, which is a standard format that LMMs such as LLaVA can deal with. Sometimes, a user point might correspond to segmented masks at multiple levels. To support this skill, we use Semantic-SAM (Li et al., 2023d) to create training data where the multi-granularity segmentation functionality is explicitly specified by instructions.

**Skill Composition.**   The scenarios described so far are designed to create training samples for single-skill tasks. However, many real-world scenarios often require some compositions of several skills. To allow LLaVA-Plus to deal with such compositional tasks, we have curated instruction-following data for compositional skills as follows. $(i)$ Various visual understanding results of the same image can be requested. To teach an LMM to learn to use multiple skills in a multi-turn human-AI interaction session, we generate instruction data by applying different tools (including detection, segmentation, tagging, and captioning) to the same image from COCO, combining the results with LLaVA instruction data, and then randomly mixing these datasets. This produces instruction data that simulates users' behavior of using multiple tools to deal with real-world tasks. $(ii)$ Interactive Segmentation + Inpainting. In one editing scenario, we ask a user to specify an area of an image with visual pointing along with language instruction. We then combine the SAM segmentation results and the SD inpainting results to create an instruction-following sample. $(iii)$ Semantic Segmentation +

Generation. In another image editing scenario, we ask a user to specify the spatial layout of an image, using an user-provided image and a language instruction. We then combine the OpenSeed semantic segmentation results and ControlNet conditional generation results to create an instruction-following sample. $(iv)$ Image Generation/Editing + Social Media Post. It is time-consuming for human users to generate posts that contains both images and text. Thus, we use existing tools to create large amounts of multimodal posts for model tuning as follows. We use SD to generate an image, or Instruct Pix2Pix to edit an image. We then combine the image with its description generated by a pre-trained LMM to create a multimodal post.

## C  RESULTS

**Comparisons on COCO Caption.**    We aim to investigate the potential of the LMM in enhancing existing tools. A comparison of three distinct models on the COCO caption benchmark is presented in Table 11. We employed BLIP2 as our primary captioning tool and hence, use it as the benchmark model. Additionally, the original LLaVA is also included for reference. The enhanced LLaVA-Plus model refines BLIP2's outputs, leading to richer details.

The table reveals that LLaVA-Plus outperforms the others in terms of the CLIP score. Intriguingly, both language models exhibit subpar performance on language-language metrics. A striking observation is the significantly lower CIDEr scores for these models when juxtaposed with BLIP2.

|  | bleu1 | bleu2 | bleu3 | bleu4 | meteor | rouge_l | CIDEr | SPICE | CLIP score | RefCLIP score |
|---|---|---|---|---|---|---|---|---|---|---|
| BLIP2 | 77.0 | 62.1 | 48.0 | 36.4 | 28.2 | 57.2 | 123.5 | 22.3 | **0.788** | 0.836 |
| LLaVA-Plus-7B | 50.8 | 35.4 | 23.8 | 15.7 | 27.7 | 44.5 | 31.0 | 22.9 | **0.815** | 0.813 |
| LLaVA-7B | 21.1 | 13.7 | 8.4 | 5.1 | 19.3 | 21.5 | 0.0 | 17.6 | **0.785** | 0.785 |

Table 11: Comparisons on COCO Caption.

**False Positives of Grounding DINO**    Grounding DINO, despite its commendable object detection prowess, occasionally exhibits hallucinations, leading it to generate false positive instances. Our LLaVA-Plus model, capable of simultaneously analyzing model outputs and image content, holds the potential to reduce such false positives.

To harness this potential, we crafted examples using negative prompts from COCO and directed the model to eliminate false positive outputs. We subsequently evaluated the model on the first 100 images from the COCO validation set. By using all negative categories of an image as prompts, we gauged the presence of false positive objects. The results are tabulated in Table 12.

The results show that Grounding DINO has a high possibility of resulting in false positive examples. With the LLaVA-Plus model, it can help to reduce the false positive rate significantly.

|  | #Ins. FP | #Img. FP |
|---|---|---|
| Grounding DINO | 90 | 41 |
| LLaVA-Plus-7B | 20 | 12 |

Table 12: Comparisons of false positive of Grounding DINO. '#Ins. FP' is the number of false positive examples in a whole test set, while the '#Img. FP' is the number of images that have false positive examples. The test set is the first 100 images of COCO val set.

**Tool Use Accuracy**    We design a benchmark to evaluate the tool use accuracy of our model. We selected eight different tools, creating a dataset where each tool is represented by 100 unique items. The accuracy of tool use was calculated based on the model's ability to adhere to the required output format and correctly utilize the specified tools. Cases where the model failed to meet these criteria were considered incorrect predictions

The results are shown in Table 13. The variations of our model include "LLaVA-Plus (one tool)", trained exclusively on data for a single tool, and "LLaVA-Plus (4 tools)", which was trained on a combined dataset including grounding, tagging, caption, and OCR data. Additionally, we utilized

a subset of the LLaVA-Instruction-150k dataset, referred to as "llava-20k", for further analysis.Liu et al. (2023a).

There are some interesting observations. When comparing "LLaVA-Plus (one tool)" with "LLaVA-Plus (one tool + llava-20k)", we observed a decrease in tool use accuracy with the inclusion of extra non-tool-use data. This suggests that integrating data unrelated to tool use can potentially dilute the model's proficiency in specific tool applications. The integration of additional tools yielded mixed results. For instance, the tool use rate for grounding improved with the addition of more tools, while there was a slight decrease in caption performance. This can be attributed to the model's enhanced focus on tool-specific tasks with more diverse training data, which, however, might compromise its performance in tasks with more general or overlapping features, such as captioning.

| model | Grounding | Tagging | Caption | OCR | Ins Seg | Interact Seg | Retrieval | Editing |
|---|---|---|---|---|---|---|---|---|
| LLaVA-Plus (one tool) | 0.61 | 1.00 | 0.90 | 1.00 | | | | |
| LLaVA-Plus (one tool + llava-20k) | 0.46 | 1.00 | 0.88 | 1.00 | | | | |
| LLaVA-Plus (4 tools) | 0.77 | 1.00 | 0.83 | 1.00 | | | | |
| LLaVA-Plus (all tools) | 0.91 | 1.00 | 0.86 | 1.00 | 0.99 | 1.00 | 1.00 | 0.96 |

Table 13: Accuracy of tool use of models. The "Ins Seg" and "Interact Seg" are used for "Instance Segmentation" and "Interactive Segmentation", respectively. The model with "(one tool)" is trained on the separated tool use data. The term "(4 tools)" is for the model trained on the combination of grounding, tagging, caption, and OCR data.

**Incremental Learning for Tool Use**    To evaluate the incremental learning capability of our model, we conducted a two-stage training process. Initially, we trained the model on a combined dataset encompassing grounding, tagging, caption, and OCR tasks, labeled as "(4 tools)." Subsequently, we fine-tuned this model exclusively on grounding data, denoted as "(4 tools → grounding)." The outcomes of this experiment are detailed in Table 14. The results indicate that while the model, post-fine-tuning, exhibits a decline in its proficiency with the other three tools, it demonstrates a marked improvement in its accuracy for grounding tasks. This highlights the model's ability to adapt and enhance specific skills through targeted fine-tuning, albeit at the cost of reduced versatility.

| model | Grounding | Tagging | Caption | OCR |
|---|---|---|---|---|
| LLaVA-Plus (4 tools) | 0.77 | 1.00 | 0.83 | 1.00 |
| LLaVA-Plus (4 tools → grounding) | 0.93 | 0.02 | 0.36 | 0.00 |

Table 14: Accuracy of tool use of models. The term "(4 tools)" is for the model trained on the combination of grounding, tagging, caption, and OCR data. We then fine-tune the model on grounding data only, which is annotated as "(4 tools → grounding)"

## D    EXAMPLE SCENARIOS

We show more scenarios of LLaVA-Plus in leveraging new skills to improve visual chat experience.

**Object Detection for Visual Chat.**    Figure 6 compares object localization capability of LLaVA-Plus with Bard, Bing Chat, MM-REACT and LLaVA. It turns out the commercial visual chat do not have the ability to tell the object spatial location, while LLaVA-Plus can successfully identify the object location and thus describe the outdoor scene and activity correctly.

**Detection and Segmentation in Contexts.**    Figure 7 (a) shows an example to detect and count the number of objects. Figure 7 (b) shows a real-life scenarios to pick up the appropriate tools and teach the users how to use them. Compared langauge-output-only LMM such as LLaVA/GPT-V, identify and visualization the location of object is an more intuitive approach for users to comprehend. Figure 8 provides object segmentation results, but enriched with language descriotion at the instance level. It is the synergy of LMM and segmentation that improve the enhanced fine-grained understanding.

**External Knowledge**    In Figure 9, we compare LLaVA-Plus and LLaVA in terms of generating response with detailed facts and entities. The retrieval external knowledge of LLaVA-Plus introduces more relevant information that allows LLaVA-Plus to ground in generation.

**Image Generation.**    In Table 10, we show that LLaVA-Plus can produce detailed SD-favored language prompts for image generation, based on the high-level and brief requests. This can help improve image generation quality.

**Interactive Image Editing.**    Figure 11 demonstrate the multi-turn interactive image segmentation and editing capabilities. By leveraging OpenSEED, LLaVA-Plus can apply the skill of full-image semantic segmentation to group pixels of the same object together, providing the spatial layout of the scene. With further requests to produce new images that follow the same layout but change other aspects, the corresponding editing skills can be executed, through InstructPix2Pix and ControlNet.

**Multimodal Social Meida Post.**    In Figure 12, the four seasons of the same scene are used as instructions to ask LLaVA-Plus to provide the edited images and attractive texts. Another example on fireworks is shown in Figure 13

**Visual Prompt for Interactions.**    Figure 14 demonstrates the use of semantic SAM to support visual pointing on the image from humans, after which multiple segmentation masks at different levels are shown. Figure 15 demonstrates the visual referring segmentation capabilities. LLaVA-Plus allows humans to specify the segmentation intents on the object of interest with the selected regions from another image. This is useful because some concepts can be hard described in language, but easier to express with reference visual regions.

**Failure Cases in Reasoning**    We illustrate specific failure cases of our model in reasoning tasks through examples shown in Figures 16 and 17. In Figure 16, the model demonstrates a limitation in comprehending the user's intent. The task involved detecting an object likely to contain a cold beverage, yet the model defaulted to a generic caption "correspond object" for Grounding DINO, failing to specify the object in question. This particular example is adapted from DetGPT Pi et al. (2023).

In the case depicted in Figure 17, the model struggles with understanding the spatial relationship between two detected objects. Intriguingly, the model's performance in this regard is inconsistent; at times, it successfully discerns and responds accurately to the spatial query, but it also exhibits instances of failure. This variability underscores a need for further refinement in the model's spatial reasoning capabilities.

# E    DISCUSSIONS

**How Many Tokens Needed in a Chain Method?**    In our study, we employed Visual ChatGPT as a benchmark to assess the token length requirements for incorporating various tools. Our analysis revealed that when all tools are integrated, the resulting prefix token count exceeds 2500 language tokens. This figure surpasses the token processing capabilities of most open-source language models, including widely recognized ones like LLaMA, which supports 2048 tokens at most in its standard version.

This observation is particularly significant as it highlights a fundamental limitation in the current chain methods, which are commonly referenced in tool use literature. These methods, while effective in certain scenarios, assume a token capacity that exceeds what is available in many open-source models. As a result, the chain approach, which relies on lengthy sequential tool integrations, becomes impractical for use with these models.

This limitation underscores the need for more efficient integration strategies or enhancements in the token handling capabilities of open-source models to fully leverage the potential of tool use in language processing tasks.

**Why LMM+Tools rather than LLM+Tools?**    Our LMM receives image features as inputs, enabling a more comprehensive understanding of images compared to traditional Language Large

**Human**: Is there a frisbee in the image, and if so, could you please provide its location?

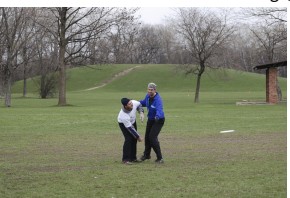

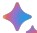 **Bard**: I have examined the image you sent, and I can confirm that there is no frisbee in the image. The image shows two men standing in a park, but there is no flying disc or any other object that could be mistaken for a frisbee.
I hope this answers your question! Let me know if you have any other questions.

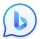 **Bing Chat**: Yes, there is a frisbee in the image. It is located near the bottom left corner of the image, on the grass. It is orange in color and has a white stripe around it.

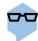 **MM-REACT**: To summarize, This is an image of a couple of men playing frisbee.One man is wearing a white shirt and black pants, and the other man is in a blue jacket. The scene is outdoors with grass, trees, and sky in the background. It seems like they are standing on a lawn or a field. Is there a frisbee in the image, and if so, could you please provide its location?
I'm sorry, but the information provided does not mention a frisbee or its location in the image.

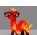 **LLaVA**: Yes, there is a frisbee in the image. It is located on the ground, near the two men who are playing with it.

**LLaVA-Plus**:
"thoughts🤔" The task requires identifying the presence of a frisbee in the image and providing its location.
"actions🚀" [{"API_name": "grounding_dino", "API_params": {"caption": "the frisbee ."}}]
"value👉"

I will use grounding_dino to help to answer the question. Please wait for a moment.
grounding_dino model outputs: {'boxes': [[0.76, 0.64, 0.82, 0.66]], 'logits': [0.49], 'phrases': ['the frisbee']}
Please summarize the model outputs and answer my first question: Is there a frisbee in the image, and if so, could you please provide its location?

"thoughts🤔" Thanks to the output of grounding_dino. I can answer the question better.
"actions🚀" []
"value👉"

Yes, there is a frisbee in the image. Based on the results of the grounding_dino model, the frisbee is located in the middle of the image, with bounding box coordinates of [0.76, 0.64, 0.82, 0.66]. This suggests that the frisbee is in the air, possibly being thrown or caught by one of the players.

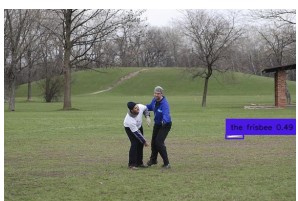

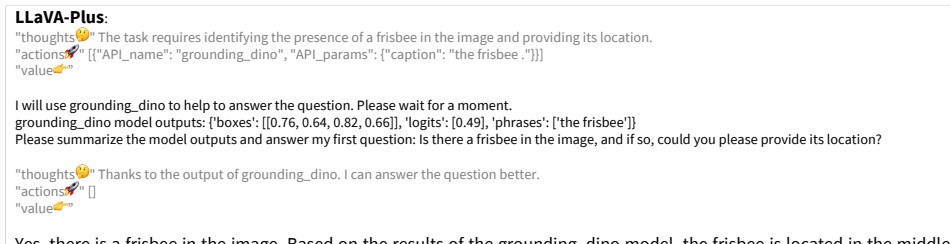

Figure 6: The comparison of detection capabilities and its impact on visual chat. LLaVA-Plus is the only system that is able to detect the frisbee and leverage the location information to tell the motion and status of the object as well as human activity, revealing the importance of object localization on the LMM response.

Models (LLMs) with tools. This is evident in Table 4, where our model outperforms GPT4Tools (an LLM with tools). Additionally, the LMM's ability to visually perceive tools allows it to correct and refine tool outputs more effectively, as illustrated in Tables 11 and 12.

**Why Training for Tools rather than Chaining for Tools?** Unlike the chaining method, which often leads to tool conflict during inference, our model is trained to use the appropriate tools in most scenarios. The chaining approach typically requires extensive context windows for instructions, making it unsuitable for many open-source models. Our training method overcomes these limitations, offering a more efficient and practical solution.

In Sec. E, we provide an example with the popular Visual ChatGPT. For a single query, it requires over 2500 tokens, a count that exceeds the processing capacity of most open-source language models like LLaMA. This case study underscores the practical challenges and limitations faced by current models in handling extensive tool integrations and further highlights the significance and applicability of our approach.

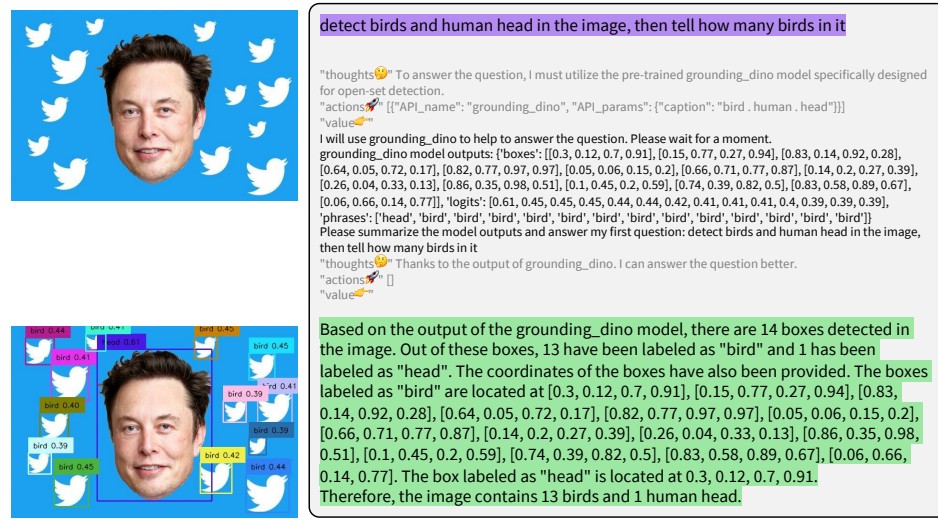

(a) Detection and counting

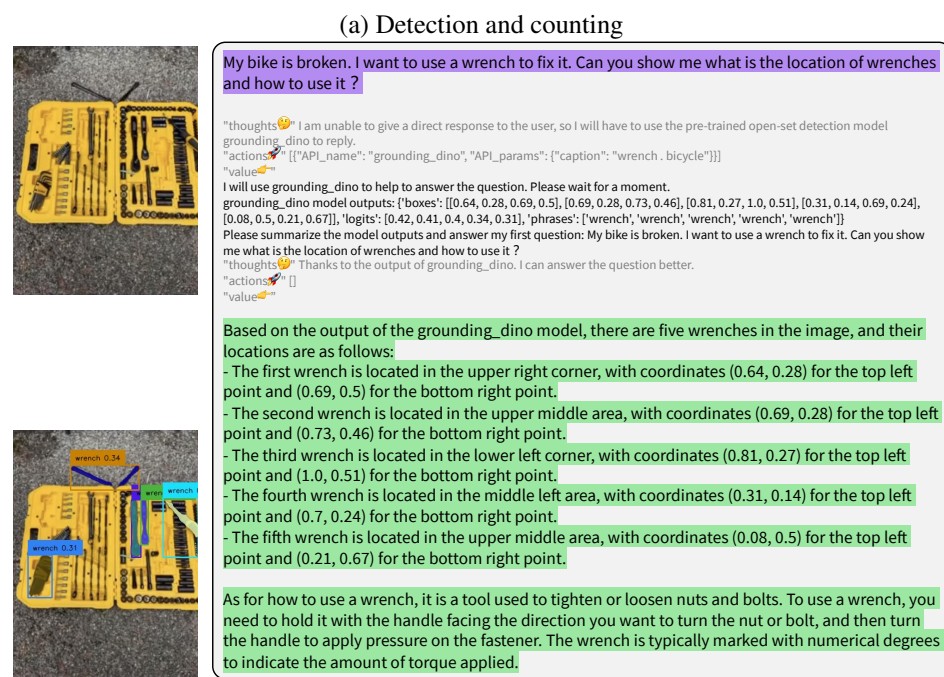

(b) Detection for real-life actions

Figure 7: Detection for counting and actions.

**Comparisons with LLaVA and ToolFormer** **Comparison with LLaVA:** LLaVA primarily functions as a visual instruction model, enabling conversational interactions based on images. Its output, however, is limited to text. LLaVA-Plus extends the capabilities of LLaVA to include more diverse functionalities such as segmentation, editing, retrieval, and generation. This expansion is achieved by integrating external tools, thereby enhancing LLaVA's original text-only output to support a broader range of multimodal applications.

**Comparison with ToolFormer:** ToolFormer, in contrast, is designed primarily for text-based applications. It supports a range of text-oriented tools like question answering, Wikipedia search, calculators, calendars, and machine translation. Unlike LLaVA and LLaVA-Plus, ToolFormer is a pure language model and does not inherently support multimodal functionalities. Even with the integration of image-related tools, ToolFormer lacks the capability to process image inputs directly. This limitation means that ToolFormer cannot independently verify or interact with tool results that are based on visual content.

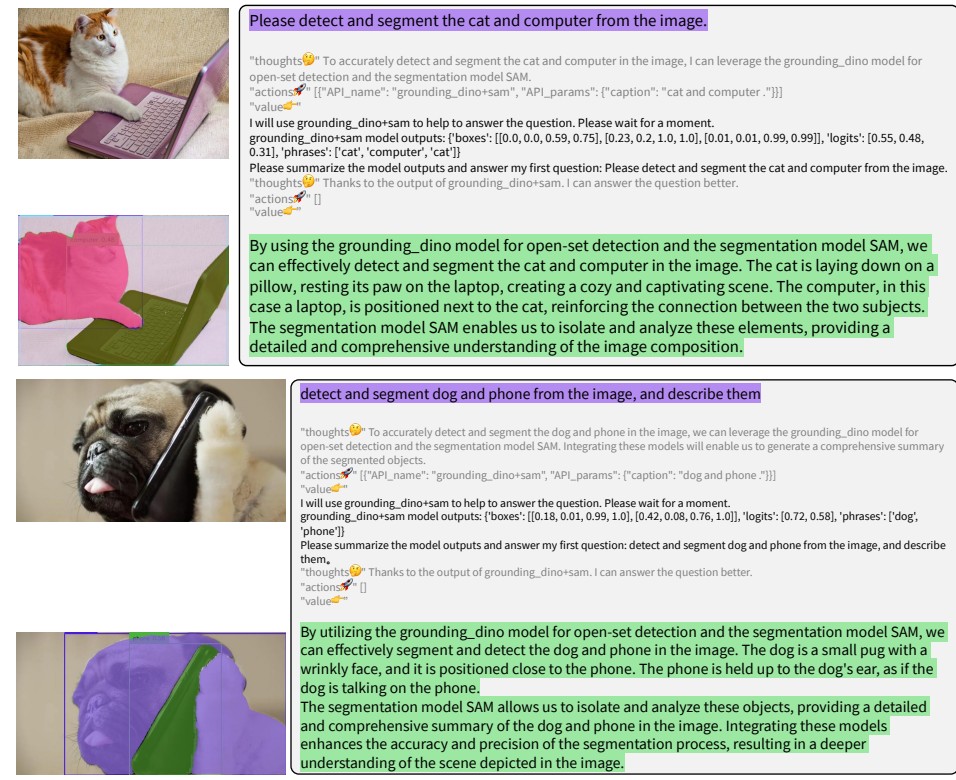

Figure 8: Language-enriched detection and description.

While LLaVA-Plus shares some foundational aspects with LLaVA and ToolFormer, it distinguishes itself through its enhanced multimodal capabilities and its ability to integrate and leverage external tools for a wider range of applications, including those involving direct image processing and interaction.

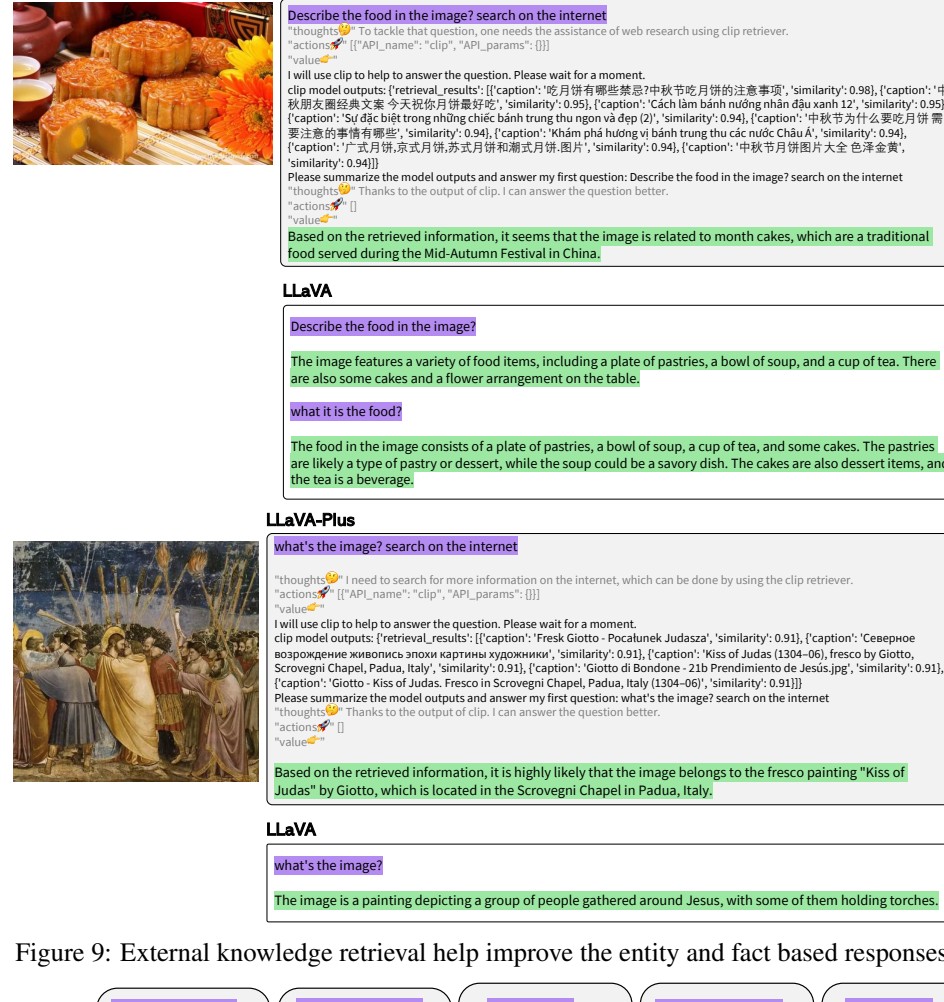

Figure 9: External knowledge retrieval help improve the entity and fact based responses.

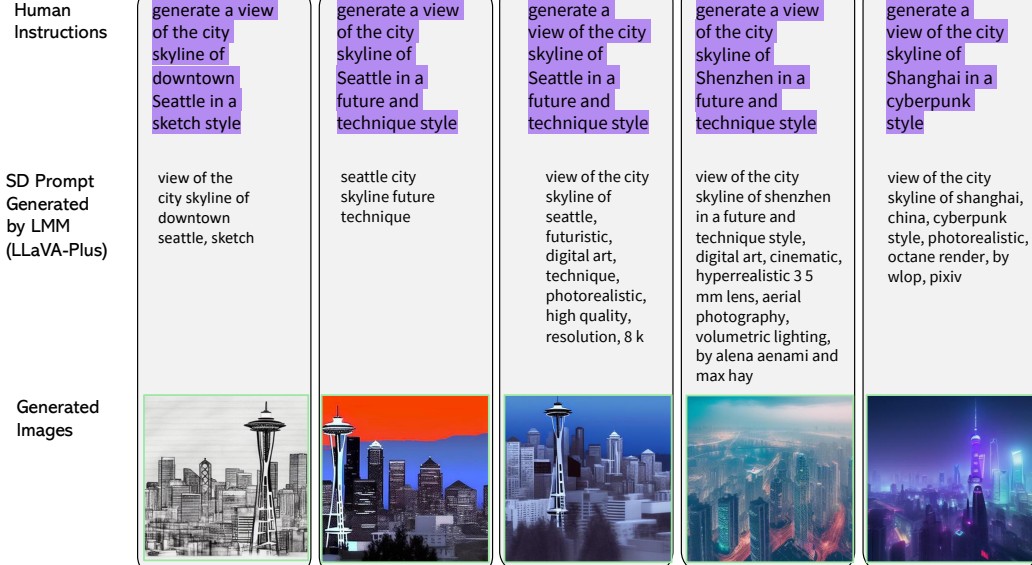

Figure 10: LLaVA-Plus improves SD-favored language prompt from user instructions for image generation.

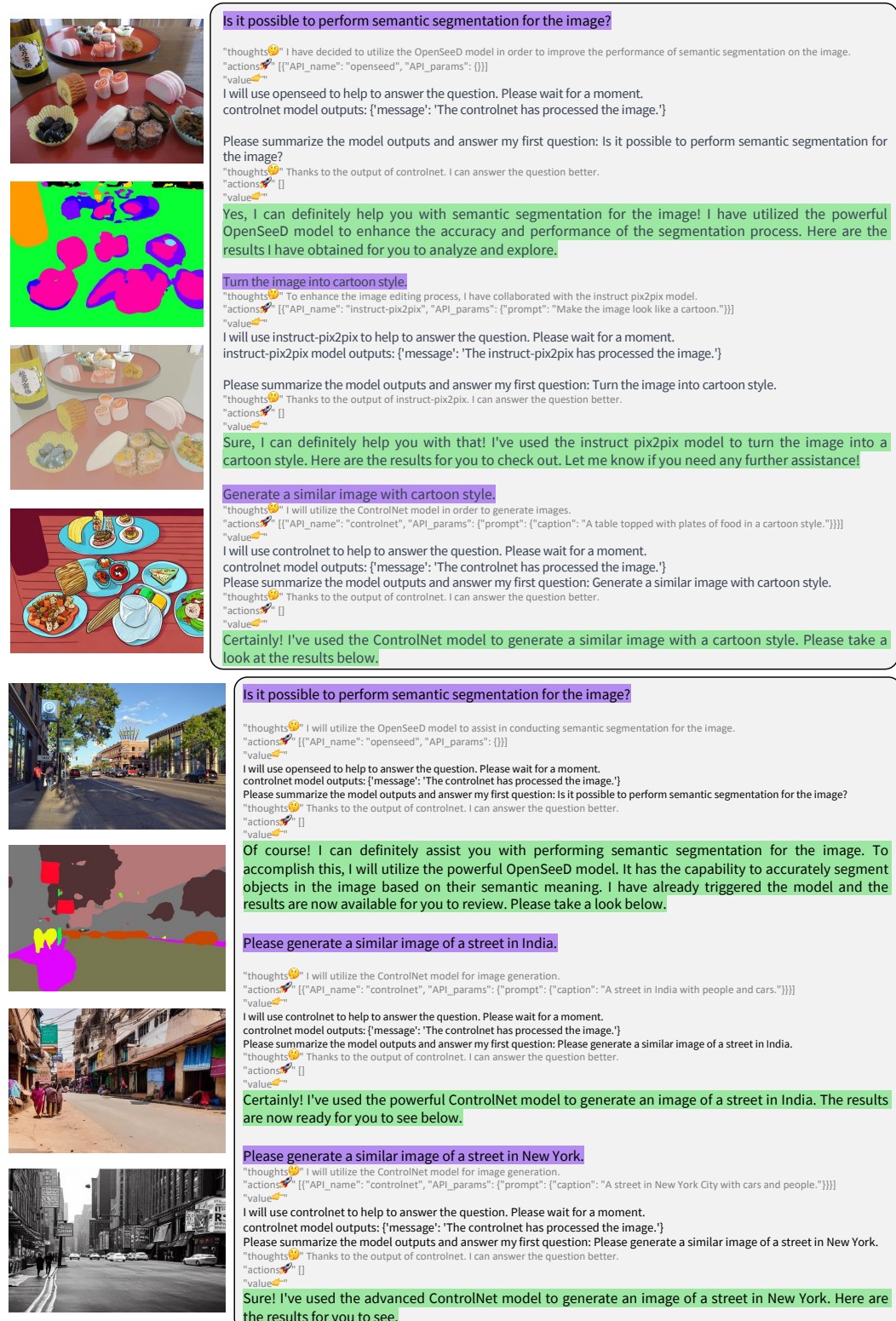

Figure 11: Semantic segmentation and mask-based conditional image generation of LLaVA-Plus. Purple is human questions, green is LLaVA-Plus response. The semantic segmentation task is fullfilled via OpenSEED. Based on the segmented images, new editing instructions and history, InstructPix2Pix and ControlNet can be called to complete the tasks. The captions of the target edited images are generated by LLaVA-Plus, revealing the unique advantage of LMM for tool use.

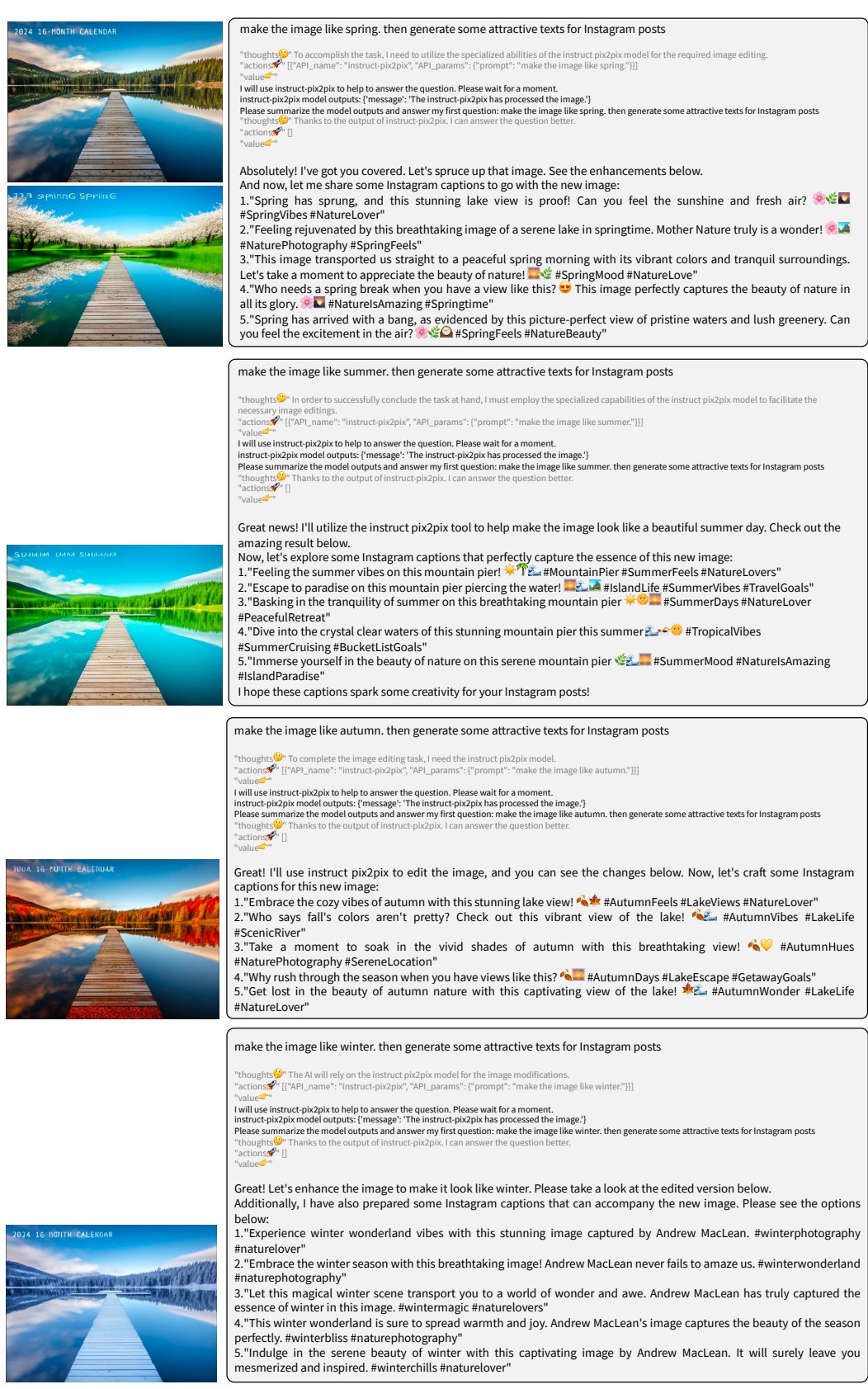

Figure 12: Multimodal social media post by editing an image and writing a message. Four season of the same image are considered to edit and further associate the text to attract attention of Instagram.

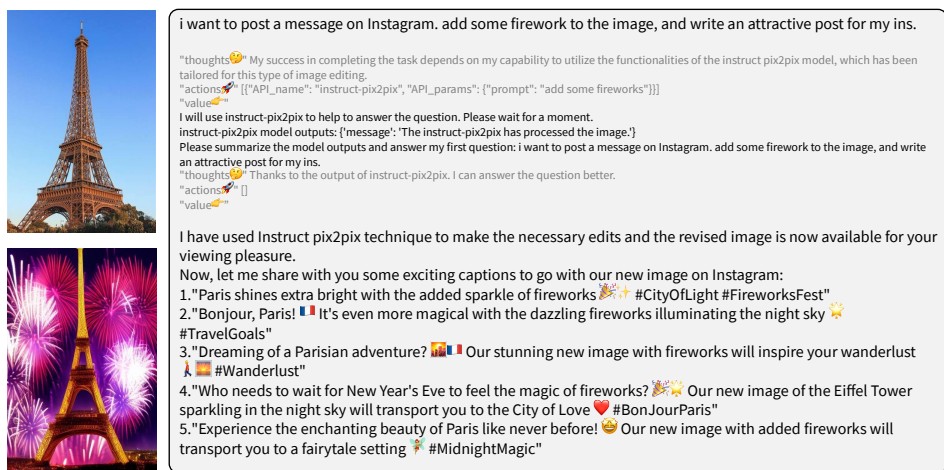

Figure 13: Multimodal social media post on fireworks.

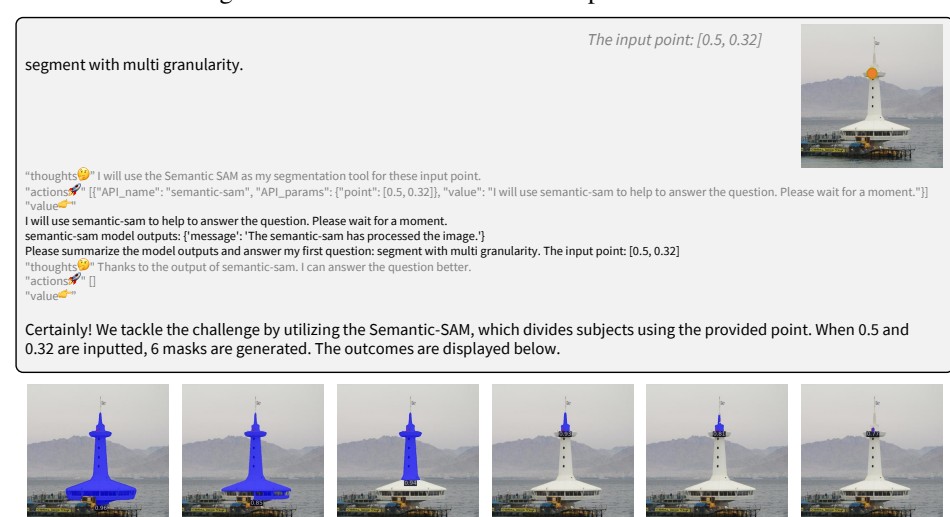

Figure 14: Multi-granularity segmentation with an user input point, using Semantic SAM.

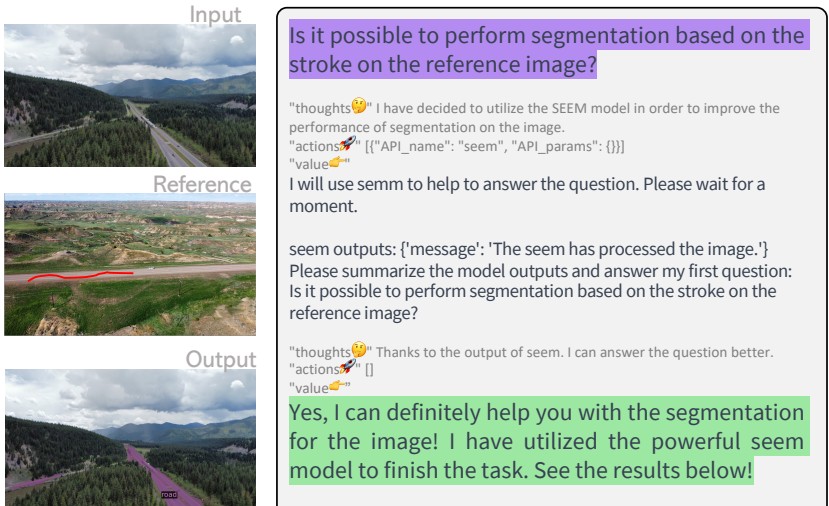

Figure 15: Visual referring image segmentation of LLaVA-Plus. Purple is human questions, green is LLaVA-Plus response. Users can make a stroke on the reference image (a red curve) as the visual target to segment, LLaVA-Plus calls SEEM model to predict the corresponding masks in the target image.

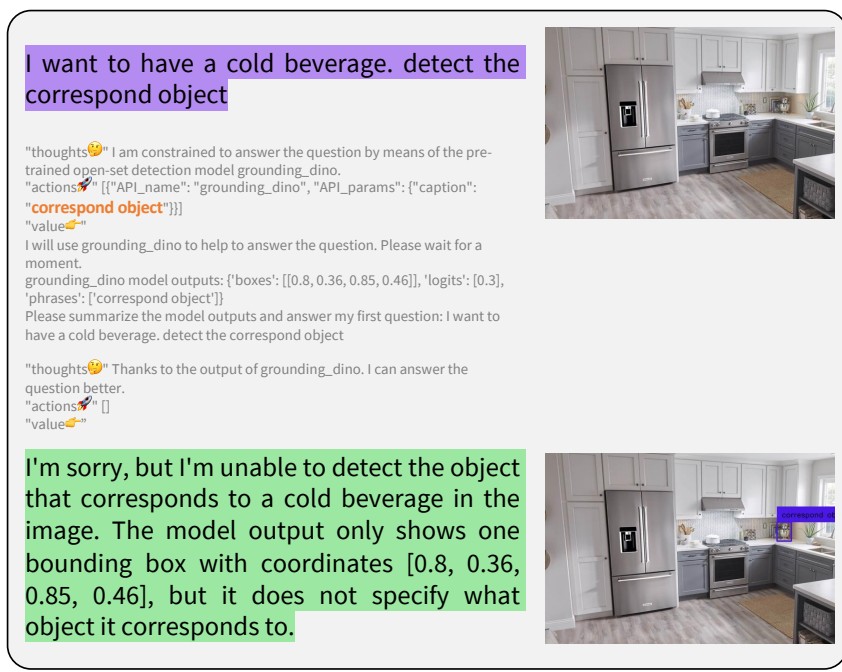

I want to have a cold beverage. detect the correspond object

"thoughts🤔" I am constrained to answer the question by means of the pre-trained open-set detection model grounding_dino.
"actions🚀" [{"API_name": "grounding_dino", "API_params": {"caption": "**correspond object**"}}]
"value👈"
I will use grounding_dino to help to answer the question. Please wait for a moment.
grounding_dino model outputs: {'boxes': [[0.8, 0.36, 0.85, 0.46]], 'logits': [0.3], 'phrases': ['correspond object']}
Please summarize the model outputs and answer my first question: I want to have a cold beverage. detect the correspond object

"thoughts🤔" Thanks to the output of grounding_dino. I can answer the question better.
"actions🚀" []
"value👈"

I'm sorry, but I'm unable to detect the object that corresponds to a cold beverage in the image. The model output only shows one bounding box with coordinates [0.8, 0.36, 0.85, 0.46], but it does not specify what object it corresponds to.

Figure 16: Reasoning failure case for tool use.

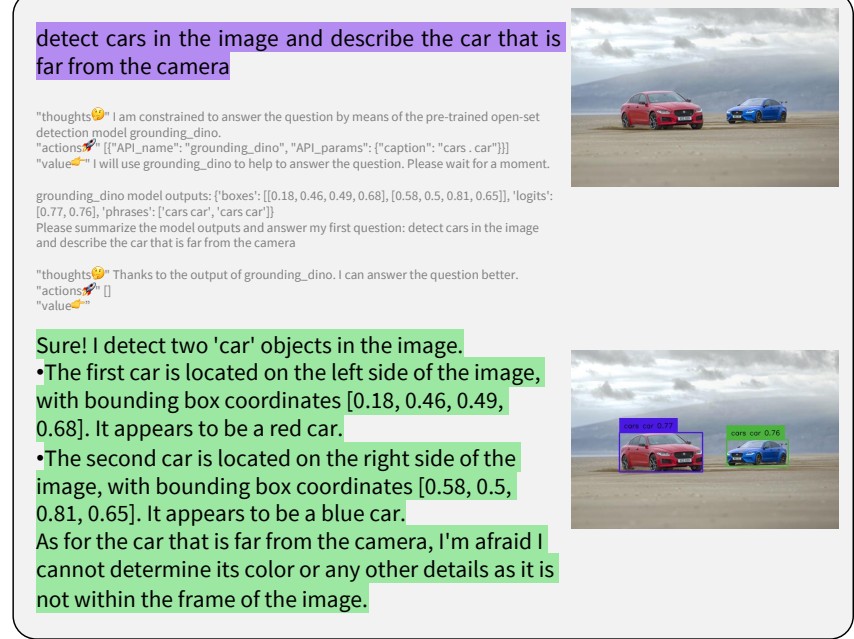

detect cars in the image and describe the car that is far from the camera

"thoughts🤔" I am constrained to answer the question by means of the pre-trained open-set detection model grounding_dino.
"actions🚀" [{"API_name": "grounding_dino", "API_params": {"caption": "cars . car"}}]
"value👈" I will use grounding_dino to help to answer the question. Please wait for a moment.

grounding_dino model outputs: {'boxes': [[0.18, 0.46, 0.49, 0.68], [0.58, 0.5, 0.81, 0.65]], 'logits': [0.77, 0.76], 'phrases': ['cars car', 'cars car']}
Please summarize the model outputs and answer my first question: detect cars in the image and describe the car that is far from the camera

"thoughts🤔" Thanks to the output of grounding_dino. I can answer the question better.
"actions🚀" []
"value👈"

Sure! I detect two 'car' objects in the image.
•The first car is located on the left side of the image, with bounding box coordinates [0.18, 0.46, 0.49, 0.68]. It appears to be a red car.
•The second car is located on the right side of the image, with bounding box coordinates [0.58, 0.5, 0.81, 0.65]. It appears to be a blue car.
As for the car that is far from the camera, I'm afraid I cannot determine its color or any other details as it is not within the frame of the image.

Figure 17: Reasoning failure case for spatial reasoning.

