# OpenReview forum: "LLaVA-Plus: Learning to Use Tools for Creating Multimodal Agents"
_ICLR.cc/2024/Conference — Submitted to ICLR 2024_

### Official Review · Reviewer_xZda · 2023-10-26

**Soundness:** 4 excellent
**Presentation:** 3 good
**Contribution:** 4 excellent
**Rating:** 10
**Confidence:** 4

**Summary:**

This paper introduces LLaVA-Plus, an end-to-end training approach aimed at enhancing large multimodal models (LMM) to create general-purpose multimodal agents. LLaVA-Plus maintains a skill repository consisting of various vision and vision-language pre-trained models, which are activated based on user instructions and input images, allowing the LMM to learn and apply skills dynamically to complete real-world tasks. The approach leverages multimodal instruction-following data for tool use and outperforms existing methods, showing superior performance in multimodal tool use and enabling new scenarios by considering query images throughout the interaction process.

**Strengths:**

This work does an incredible job of combining tool chaining methods with visual and language models that figures out what tools to use given a task. This method has been thoroughly tested with a large number of tasks and datasets and has been shown to b

**Weaknesses:**

My only gripe with the paper is the presentation. Some parts of the paper have been written in such a way that does not give the impression that this work is sufficiently novel from LLaVa and ToolFormer.

**Questions:**

Here are some suggestions to improve the presentation of the work:
- In the discussion section, include the following subsections `How does our work differ from LLaVA?` and `How does our work differ from ToolFormer?`
- Figure 2 should be expanded into a bigger pipeline, which shows the different skills in the skill repository. It is hard to tell which part is the contribution of the paper, so emphasizing which modules are added by the paper will help highlight that.
- Adding a small introduction of the LLaVA paper would help the readers not go back and forth between the two papers to figure out the differences. The authors could consider adding that to a Preliminaries section.

---

> ### Author Response · Authors · 2023-11-23
> **Responses to Reviewer #xZda**
>
> Thanks for your valuable comments. We first have brief summary of your concerns and our responses, followed by detailed responses.
>
> ## Brief Summarization:
>
> 1. **Suggestions to discuss related works**
>
> Response: We added a new section to compare LLaVA-Plus with LLaVA and ToolFormer in Sec. E of the revised paper.
>
> 2. **Suggestions to improve Figure 2.**
>
> Response: We updated Figure 2 with more detailed skill repos as you suggested.
>
> 3. **Writing suggestions about the preliminaries of LLaVA**
>
> Response: We added a new section to provide a comprehensive background on Visual Instruction Tuning in LLaVA.
>
> ## Detailed Responses:
>
> - **W1**: In the discussion section, include the following subsections How does our work differ from LLaVA? and How does our work differ from ToolFormer?
>
> **Response**: Thank you for the suggestion to delineate how our work, LLaVA-Plus, differs from both LLaVA and ToolFormer. We have included a new section at the Sec. E in our paper to comprehensively address these comparisons.
>
> *Comparison with LLaVA:*
>
> - LLaVA primarily supports conversational interactions based on images. Its output, however, is limited to text.
> - LLaVA-Plus extends the capabilities of LLaVA to include more diverse functionalities such as segmentation, editing, retrieval, and generation. This expansion is achieved by integrating external tools, thereby enhancing LLaVA's original text-only output to support a broader range of multimodal applications.
>
> *Comparison with ToolFormer:*
>
> - ToolFormer, in contrast, is designed primarily for text-based applications. It supports a range of text-oriented tools like question answering, Wikipedia search, calculators, calendars, and machine translation.
>
> - Unlike LLaVA and LLaVA-Plus, ToolFormer is a pure language model and does not inherently support multimodal functionalities. Even with the integration of image-related tools, ToolFormer lacks the capability to process image inputs directly. This limitation means that ToolFormer cannot independently verify or interact with tool results that are based on visual content.
>
> While LLaVA-Plus shares some foundational aspects with LLaVA and ToolFormer, it distinguishes itself through its enhanced multimodal capabilities and its ability to integrate and leverage external tools for a wider range of applications, including those involving direct image processing and interaction.
>
> - **W2**: Figure 2 should be expanded into a bigger pipeline, which shows the different skills in the skill repository. It is hard to tell which part is the contribution of the paper, so emphasizing which modules are added by the paper will help highlight that.
>
> **Response**: Thank you for your valuable suggestion to enhance the clarity and detail of Figure 2 in our paper. We have taken your feedback into account and made significant modifications to the figure.
>
> We have expanded Figure 2 to include a more detailed depiction of the various skills within our skill repository. This expanded view provides a comprehensive understanding of how each skill is integrated into the overall pipeline of LLaVA-Plus.
> The primary contributions of our paper are now clearly highlighted in shaded regions within the figure. These areas specifically illustrate the extensions and enhancements that our work, LLaVA-Plus, brings to the existing LLaVA model.
>
> In LLaVA, the process was limited to Step 1 (instruction intake) and Step 4 (response generation). In contrast, LLaVA-Plus significantly expands this pipeline, introducing intermediate steps that enhance the model's ability to process and respond to inputs more effectively. These additions are clearly marked to showcase the advancements we have made.
>
> We believe that these modifications will effectively demonstrate the enhancements introduced by LLaVA-Plus, making it easier for readers to identify and appreciate the novel aspects of our work. Thanks again for your valuable suggestions.
>
>
> - **W3**: Adding a small introduction of the LLaVA paper would help the readers not go back and forth between the two papers to figure out the differences. The authors could consider adding that to a Preliminaries section.
>
> **Response**:  Thank you for your valuable suggestion to provide context for our work in relation to the LLaVA paper. In response, we have introduced a new section in our paper titled "Preliminaries: Visual Instruction Tuning in LLaVA" (Section 2.1). This section is dedicated to offering a concise overview of the LLaVA model, aiming to familiarize readers with its foundational concepts and functionalities.

---

### Official Review · Reviewer_aMGS · 2023-10-30

**Soundness:** 2 fair
**Presentation:** 2 fair
**Contribution:** 2 fair
**Rating:** 3
**Confidence:** 4

**Summary:**

The focus of paper is on a general-purpose multimodal assistant, LLaVA-Plus, which is based on a range of large vision and vision-language pertained models. LLaVA-Plus is trained on multimodal instruction-following data. The data can be used for visual understanding, visual generation, external knowledge retrieval and compositions thereof. The approach is based on a combination of tool chaining with LLMs and end-to-end training with LLMs. LLaVA-Plus uses images during human-AI interaction to improve LLM’s planning and reasoning ability. LLaVA-Plus is based on an the LLaVA model, but doesn’t only involve performing user-oriented dialogues, and also performs skill-oriented dialogues. For this purpose, agent calls different tools like various vision and large language models to execute a task. The way LLaVa-Plus works follows: users provide a a task description with a designated image. Given the provided inputs, the agent chooses the tool and writes the appropriate prompt as the tool argument. The assistant outputs the answer to humans. The authors present the results of their approach on multiple benchmarks including VisiT-Bench with different tasks.

**Strengths:**

I like the approach using the open-weight LLaVA model, which provides a lot more ground for actual meaningful experimentation on the model itself than closed proprietary models like GPT-4.  The authors propose a fairly straightforward augmentation to the LLaVA model that on the surface appears to provide an ability to expand its vocabulary of skills by determining which external models to call on for a given task, using visual-linguistic supervision in an AutoGPT-like approach.  The paper is also written pretty clearly with only a few typos or disfluencies that impede understanding.

**Weaknesses:**

Much of the meaningful technical information, such as substantive results, are limited to the appendix.  This makes me question the soundness and what the contribution of the paper actually is. We don't even get to the related work until page 6 and so the experiment and results are squeezed at the very end.

The word "planning" is used but the paper doesn't really have anything to do with planning as commonly understood.  There is no goal to be trained to achieve. The order of which API should be called first and what goes next shouldn’t be confused with planning approaches.  In addition, the use of "tools" is misleading. This leads the reader (at least it read me) to think this was about robotic planning and object-affordance exploitation, but instead this is more about using the appropriate API call to different systems in an AutoGPT-style approach.  It is not clear to me how this approach leads to a general multimodal agent.  This was submitted to the "representation learning for computer vision, audio, language, and other modalities" but there's nothing about representation learning or conceptual understanding here.  All the model does is call upon other unaltered submodels to perform vision-language tasks.

The approach seems not to be fully generalizable as there appears to be little error checking and validation of the outputs of the individual sub-models invoked, and therefore errors could propagate downstream and compound.  The authors mention in the conclusion that LLaVA-Plus still suffers from hallucination and tool conflict.  At the very least the tool conflict problem would need to be addressed to drive home the contribution, because otherwise such a plug-and-play architecture is unlikely to generate useful outputs more frequently than a manual chaining together of the individual sub-models.

Some other points:

"develop general-purpose assistants for natural language tasks have been proved effective" - “Proved effective” is an overstatement.  There are still plenty of reasoning domains (e.g., logic, physical reasoning problems) where ChatGPT and related models at the very least are unreliable.  This point does not negate the central thrust of this paper, but it should be softened.

"the combinations of these tools lead to emergent abilities that show signs of higher intelligence" - Are the authors simply mentioning the Society of Mind approach here, or do they claim that LLaVA-Plus also displays signs of higher intelligence?  Extraordinary claims required extraordinary evidence.

A Full Dialogue of LLaVA-Plus (Figure 2) - This doesn’t seem like a full dialogue session.  It’s a diagram.  I would expect to see an actual example at each step along with a representation of the relevant information flow throughout the process.

**Questions:**

1) In Figure 3 top right, how would one confirm there is a sea in that image? It looks more like the image generated is of a bike resting against a blue wall.  In other words the prompt didn’t give you what you seemed to want.

2) How did you approach your problem rewriting the questions using GPT4 and making sure output is what you want? How would you resolve the conflict that might be across the tools?

3) What is the objective function?  At the very high level, an auto-regressive objective has been mentioned. Assuming the reader is familiar with this concept, please include the details, as there is an assistant and feedback loops in your approach.

4) Does expanding to new skills degrade LLaVA-Plus's abilities in previous skills?

5) None of the external models leveraged here is perfect.  Can LLaVA-Plus do any kind of error correction or validation to make sure errors from sub-models don’t propagate downstream?

6) "We follow the self-instruct to curate the data by using GPT-4 as the labeler." - I do not understand this sentence (ungrammatical)

7) How much data curation is required for each skill?  This seems like an intensive process and a bottleneck in scaling up.

8) "For Xq, we use GPT-4 to write a set of instructions that require the use of tools for proper answers." - It’s a shame the authors apparently have to turn to the proprietary GPT-4 for this. Was something like plain LLaVA not sufficient?

---

> ### Author Response · Authors · 2023-11-23
> **Responses to Reviewer #aMGS (part 1)**
>
> Thanks for your valuable comments. We first have brief summary of your concerns and our responses, followed by detailed responses.
>
> ## Brief Summarization:
>
> 1. **Concerns about the use of "planning" and "tools".**
>
>     R: We show that we follow the practice of previous works like Visual ChatGPT, AutoGPT, and ToolFormer to use these terms. We updated the paper with extra footnotes to clarify the use of the terms.
>
> 2. **Concerns about the LMM+Tools+Train designs of our framework.**
>
>     R: We show that our choice is the most reasonable way to improve LMM, i.e., LMM + Tools + Instruction Tuning, avoiding the long context window as in prompt based methods,  enabling image context in chain ways and enabling tool result aggregation. We added a new paragraph in our revised paper to discuss the design in Sec. E.
>
> 3. **Writing suggestions about some sentences and figures.**
>
>     R: We have updated these following your suggestions. Thanks for pointing out these problems.
>
> ## Detailed Responses:
>
> - **W1**: Much of the meaningful technical information, such as substantive results, are limited to the appendix. This makes me question the soundness and what the contribution of the paper actually is. We don't even get to the related work until page 6 and so the experiment and results are squeezed at the very end.
>
> **Response**: Thank you for your feedback regarding the organization and presentation of technical information in our paper. We have chosen to present our method before delving into related work, as this structure allows us to establish a clear foundation of our approach, which is crucial for understanding the subsequent discussions and comparisons.
>
> We are aware of the upcoming expansion of page limitations to ten pages for the ICLR camera-ready version. We intend to utilize this additional space to expand on our experiments and results, ensuring that these critical aspects of our paper are given comprehensive coverage.
>
> - **W2**: The word "planning" is used but the paper doesn't really have anything to do with planning as commonly understood. There is no goal to be trained to achieve. The order of which API should be called first and what goes next shouldn’t be confused with planning approaches. In addition, the use of "tools" is misleading. This leads the reader (at least it read me) to think this was about robotic planning and object-affordance exploitation, but instead this is more about using the appropriate API call to different systems in an AutoGPT-style approach. It is not clear to me how this approach leads to a general multimodal agent. This was submitted to the "representation learning for computer vision, audio, language, and other modalities" but there's nothing about representation learning or conceptual understanding here. All the model does is call upon other unaltered submodels to perform vision-language tasks.
>
> **Response**: Thanks for the question.
>
> *Clarification on the Use of “Planning”*: In the context of our paper, “planning” refers to the process by which the model determines the most appropriate tool to use for a given task. This usage aligns with similar terminologies employed in projects like HuggingGPT and AutoGPT. To prevent any ambiguity and ensure clarity, we have added a footnote in our paper that explicitly defines our use of “planning” in this specific context.
>
> *Usage of “Tools” in the Paper*: The term “tools” in our research is used to describe the APIs and systems that our model interfaces with to perform various tasks. This terminology is consistent with its usage in notable works such as ToolFormer and Visual ChatGPT. We appreciate your observation regarding the potential for misunderstanding and have added explanatory notes in the paper to clarify this terminology.
>
> *Field of Submission and Representation Learning*: Regarding the field of submission for our paper, we carefully considered the categories listed on the ICLR website and selected "representation learning for computer vision, audio, language, and other modalities" as the most fitting for our work. This choice reflects the multimodal nature of our model and its applicability across various domains. However, we are open to suggestions and willing to reconsider the category if a more suitable one is recommended.

---

> ### Author Response · Authors · 2023-11-23
> **Responses to Reviewer #aMGS (part 2)**
>
> - **W3**: The approach seems not to be fully generalizable as there appears to be little error checking and validation of the outputs of the individual sub-models invoked, and therefore errors could propagate downstream and compound. The authors mention in the conclusion that LLaVA-Plus still suffers from hallucination and tool conflict. At the very least the tool conflict problem would need to be addressed to drive home the contribution, because otherwise such a plug-and-play architecture is unlikely to generate useful outputs more frequently than a manual chaining together of the individual sub-models.
>
>  **Responses**: Thank you for your insightful questions, which touch upon critical aspects of our approach: the use of Large Multimodal Models (LMM) for tool use and the decision to train the model for this purpose.
>
> *LMM vs. LLM for Tool Use*: Our choice to utilize LMMs, as opposed to Language Large Models (LLMs), is grounded in the unique capability of LMMs to process image features. This ability allows for a more comprehensive understanding of visual information, which is crucial in our context. As evidenced in Table 4(a), our LMM-based model demonstrates superior performance compared to GPT4Tools, which employs LLMs. The visual comprehension ability of LMMs significantly aids in the correction and refinement of tool outputs, a key advantage detailed in Tables 11 and 12. This feature plays a vital role in mitigating errors and enhancing the reliability of our model.
>
> *Training vs. Chaining for Tool Use*: Regarding the training methodology, we opted for a trained model approach over the more common chaining method. Chaining often leads to tool conflicts during inference, a problem that our trained model approach successfully mitigates. As demonstrated our paper, our model is adept at selecting and using the appropriate tools in most scenarios, thus reducing the likelihood of error propagation and compounding.
>
> Thanks again for your suggestions. We have added a new section to discuss the motivation and advantages of our design in Sec. E. A detailed discussion about of motivations is available in **Responses to Common Concerns**.
>
>
> - **W4**: "develop general-purpose assistants for natural language tasks have been proved effective" - “Proved effective” is an overstatement. There are still plenty of reasoning domains (e.g., logic, physical reasoning problems) where ChatGPT and related models at the very least are unreliable. This point does not negate the central thrust of this paper, but it should be softened.
>
> **Responses**: Thanks for your suggestions. We have updated the paper follfollowing your suggestions. We chenged the sentence to “develop general-purpose assistants for natural language tasks have been proved effective in many tasks” to constraint its board.
>
> - **W5**: "the combinations of these tools lead to emergent abilities that show signs of higher intelligence" - Are the authors simply mentioning the Society of Mind approach here, or do they claim that LLaVA-Plus also displays signs of higher intelligence? Extraordinary claims required extraordinary evidence.
>
> **Responses**: Thanks for the question. We would like to show that LLaVA-Plus displays the capabilities to more complex tasks like segmentation, editing, and retrieval. We cite the Society of Mind approach as the high-level idea is shared. Our model aggregate multiple tool outputs, which is similar with the Society of Mind.

---

> ### Author Response · Authors · 2023-11-23
> **Responses to Reviewer #aMGS (part 3)**
>
> - **W6**: A Full Dialogue of LLaVA-Plus (Figure 2) - This doesn’t seem like a full dialogue session. It’s a diagram. I would expect to see an actual example at each step along with a representation of the relevant information flow throughout the process.
>
> **Response**: Thank you for your question regarding the depiction of a full dialogue session in LLaVA-Plus as illustrated in Figure 2.
>
> *Clarifying the Dialogue Flow in Figure 2 and Table 1*: To provide a comprehensive example of a dialogue session, we have detailed an actual dialogue in Table 1, which correlates directly with the steps outlined in Figure 2. Here's how the dialogue in Table 1 aligns with the process steps in Figure 2:
>
>   - Step 1 - Initial Query: The first human query presented in Table 1 corresponds to Step 1 in Figure 2. This step initiates the dialogue and sets the context for the interaction.
> - Step 2 - Tool Invocation: Following the initial query, the assistant’s action of invoking a specific tool, as detailed in the second paragraph of Table 1, aligns with Step 2 in Figure 2. This demonstrates the assistant's decision-making process in selecting an appropriate tool for the query.
> - Step 3 - Tool Output: The second human query in Table 1 represents the tool output, correlating with Step 3 in Figure 2. This step showcases the output provided by the invoked tool in response to the initial query.
> - Step 4 - Final Response: Lastly, the concluding response from the assistant, as depicted in the final paragraph of Table 1, corresponds to Step 4 in Figure 2. This step illustrates the assistant’s final response to the human query after processing the tool output.
>
> The dialogue example in Table 1 is designed to provide a real-world representation of the information flow and interaction process depicted abstractly in Figure 2. This example aims to clarify the practical application and functionality of LLaVA-Plus in a full dialogue scenario.
>
> - **Q1**: In Figure 3 top right, how would one confirm there is a sea in that image? It looks more like the image generated is of a bike resting against a blue wall. In other words the prompt didn’t give you what you seemed to want.
>
> **Response**: Thank you for your question regarding the interpretation of the image depicted in the top right of Figure 3.
> The image in question shows a bike leaning against a bench, set against a background that may initially seem ambiguous. On closer inspection, elements resembling a sunken boat are visible, and the presence of wave-like patterns hints at an underwater or under-sea setting. This interpretation aligns with the intended representation of a sea environment, although we acknowledge that the portrayal might not be immediately evident.
>
> *Action to Enhance Clarity*: To address this ambiguity and ensure the image aligns more closely with the intended prompt, we have regenerated the image using our pipeline. The updated image, now included in Figure 3, provides a clearer depiction of the sea environment that we initially sought to convey.
>
> Thanks again for your valuable suggestions, which help to improve our paper a lot.

---

> ### Author Response · Authors · 2023-11-23
> **Responses to Reviewer #aMGS (part 4)**
>
> - **Q2**: How did you approach your problem rewriting the questions using GPT4 and making sure output is what you want? How would you resolve the conflict that might be across the tools?
>
> **Response**: Thank you for your insightful questions regarding our approach to question rewriting and conflict resolution in tool use. In our model, each skill is associated with a specific tool, effectively eliminating the possibility of tool conflicts within the same skill. However, conflicts can arise when dealing with different skills due to the inherent ambiguities in human instruction. For instance, a grounding task might yield varying outputs like a simple box or a box with an accompanying mask.
>
> To mitigate such conflicts, our data processing pipeline focuses on two main strategies: we try to avoid ambiguous questions that could leads to conflicts when constructing instruction data. There are two approaches we used during our data processing pipeline.
>
> - Diversifying Questions: We intentionally diversify the questions in our dataset to reduce ambiguities and improve tool use accuracy. This approach helps in clearly defining the task for each tool, thereby reducing the likelihood of conflicting interpretations.
> - Quality Control in Data Generation: Our data is primarily generated using GPT4V, which, while powerful, can produce hallucinations or undesirable data items. To ensure data quality, we apply heuristic methods to filter out low-quality data:
>     - *Removal of Apologetic Responses*: We have filtered out responses containing phrases like “I’m sorry”, which often indicate uncertainty or errors in model-generated content.
>     - *Exclusion of Language Model Content*: Entries containing references to “AI language model” or “language AI model” have been removed since we plan to train a multimodal model rather than a pure language model.
>     - *Validation of CLIP-Retrieval Data*: For data pertaining to CLIP-Retrieval, we ensured that only items containing correct answers, as verified against ground-truth labels, were retained. Our selection criteria involve keeping items that match any of the listed ground truths.
>     - *Accuracy in Object Detection and Segmentation Data*: We have excluded data where the provided answers contain bounding boxes that do not align with the established ground truths, ensuring accuracy in object detection and segmentation tasks.
>
> By implementing these measures, we aim to minimize conflicts and ambiguities, ensuring that the model produces reliable and accurate outputs in line with the intended instructions.
>
> - **Q3**: What is the objective function? At the very high level, an auto-regressive objective has been mentioned. Assuming the reader is familiar with this concept, please include the details, as there is an assistant and feedback loops in your approach.
>
> **Response**: Thank you for your question regarding the objective function in our model. To clarify, our approach involves constructing data in a text format suitable for training an auto-regressive model. This model is designed to learn how to accurately answer user questions.
>
> *Data Representation and Training Approach*:
>
>    - We represent each interaction as a single sequence in our dataset, which is exemplified in Table 1. This sequence encapsulates the entire dialogue flow, including both the human queries and the assistant's responses.
>    - The format of the data, as illustrated in Equation (2), is structured as follows:
>     $Human : I_q < n> X_q<STOP> Assistant : X_{skill\underline{} use}<STOP>$
>     $Human : X_{skill\underline{} result}<STOP> Assistant : X_{anwser}<STOP>$
>    - In this representation, $I_q$ and $X_q$ denote the initial query and its context, $X_{skill\underline{}use}$ represents the assistant's tool usage, $X_{skill\underline{}result}$ is the outcome of the tool use, and $X_{answer}$ is the assistant's final response.
>
> *Objective Function and Loss Calculation*
>
>    - The loss is specifically computed on the $X_{skilluse}<STOP>$ and $X_{answer}<STOP>$ tokens. This approach ensures that the model is trained to predict these critical tokens accurately, which are essential for effective tool use and generating accurate responses.
>    - By focusing the loss calculation on these parts of the sequence, our objective function aligns the model's learning with the key task of providing relevant and accurate tool-based assistance in response to user queries.
>
> Our objective function is tailored to facilitate the learning of an effective dialogue flow, emphasizing the correct usage of tools and the generation of appropriate responses, thereby enabling our auto-regressive model to function effectively within the assistant and feedback loop framework.

---

> ### Author Response · Authors · 2023-11-23
> **Responses to Reviewer #aMGS (part 5)**
>
> - **Q4**: Does expanding to new skills degrade LLaVA-Plus's abilities in previous skills?
>
> **Response**: Thanks for your insightful question. We added two more experiments to see if more tools will degrade the tool use rate. The integration of additional tools yielded mixed results. For instance, the tool use rate for grounding improved with the addition of more tools, while there was a slight decrease in caption performance. This can be attributed to the model's enhanced focus on tool-specific tasks with more diverse training data, which, however, might compromise its performance in tasks with more general or overlapping features, such as captioning.
> A detailed result is available in Table 13.
>
> - **Q5**: None of the external models leveraged here is perfect. Can LLaVA-Plus do any kind of error correction or validation to make sure errors from sub-models don’t propagate downstream?
>
>
> **Response**: Thanks for your insightful question. Theoretically, we can do the error correction, since we use the LLaVA which is a Large Multimodal Model (LMM). This stems from its ability to process and interpret image inputs directly. The visual information provided by these inputs can play a crucial role in enabling the model to identify and rectify errors or inconsistencies in the outputs of sub-models. A practical instance of this error correction ability is demonstrated in Table 12, focusing on the Grounding DINO tool. We observed that the original Grounding DINO model occasionally produces false positive predictions. However, LLaVA-Plus has shown the capability to filter out these inaccurate predictions effectively. This results in enhanced accuracy and reliability, as the model leverages its multimodal understanding to validate and correct the tool outputs.
>
> - **Q6**: "We follow the self-instruct to curate the data by using GPT-4 as the labeler." - I do not understand this sentence (ungrammatical)
>
> **Response**:  Thank you for pointing out the need for clarity in our description. We have revised the sentence to more accurately convey our methodology: "In alignment with the LLaVA approach, we input image information into a text-only GPT-4 model, prompting it to generate both questions and responses based on the visual data."
>
> - **Q7**: How much data curation is required for each skill? This seems like an intensive process and a bottleneck in scaling up.
>
> **Response**: Thank you for your question regarding the data curation process for each skill in our model. We have found through practical experimentation that a dataset comprising approximately 5000 samples is generally sufficient to effectively train the model in invoking the necessary tools for a given skill. This quantity of data strikes a balance between comprehensiveness and manageability, ensuring that the model learns to accurately utilize the tools while keeping the curation process feasible.

---

> ### Author Response · Authors · 2023-11-23
> **Responses to Reviewer #aMGS (part 6)**
>
> -  **Q8**: "For Xq, we use GPT-4 to write a set of instructions that require the use of tools for proper answers." - It’s a shame the authors apparently have to turn to the proprietary GPT-4 for this. Was something like plain LLaVA not sufficient?
>
> **Response**:  Thanks for the question. LLaVA, being a Large Multimodal Model (LMM), involves more complexity and cost compared to a text-only model like GPT-4. Our goal was to utilize a model that could efficiently prompt and build data.
>
> In our preliminary study, we find that the capability of the teacher is crucial to the quality of the generated instruction-following data (L128-L130). Until the submission deadline, the largest Vicuna model was 13B. Just as the reviewer’s concern, its complex reasoning and spatial reasoning capability is still limited and is behind proprietary models including ChatGPT and GPT-4. However, the recently released LLaMA-2-70B-Chat appears to have narrowed the gap. Due to the large size of the model, it requires a huge amount of VRAM and has a slow inference speed. We conducted a preliminary study on around 200 samples for each category.
>
> Specifically, we generate 200 samples for each category (conversation, detailed description, complex reasoning), using LLaMA-2-70B-Chat, ChatGPT, and GPT-4.
>
> After generating the response, we find that unlike previous open-source models, LLaMA-2-70B-Chat can start to follow complex instructions like creating multimodal instructions.
>
> However, it still fails in the conversation category, as we find that the LLaMA-2-70B-Chat is not correctly following the conversation format. This may be potentially fixed with more sophisticated prompt tuning. However, due to the limited rebuttal period, we do not evaluate on the conversation category. This is also one of the main limitations we find of LLaMA-2-70B-Chat.
>
> We then quantitatively evaluate the generated instructions using GPT-4 as the judge: (1) the correctness of the answers generated, and (2) the complexity of the instructions generated for complex reasoning questions.
>
> |  | Correctness | Complexity |
> | --- | --- | --- |
> | LLaMA-2-70B-Chat | 8.7 | 7.4 |
> | ChatGPT | 9.5 | 9.2 |
>
> These initial results are promising, and suggest that our pipeline can be potentially applied to open-source models as their capabilities are improved. We look for more comprehensive studies and deeper explorations for future research.

---

### Official Review · Reviewer_3S2M · 2023-11-01

**Soundness:** 2 fair
**Presentation:** 3 good
**Contribution:** 2 fair
**Rating:** 6
**Confidence:** 3

**Summary:**

The authors key contribution is  instruction-tuning for large multimodal models (LMM) to use diverse tools. Per the authors -- compared with tool-augmented LLMs, LLaVA-Plus is distinct in that the image query is directly grounded in and actively engaged throughout the entire human-AI interaction sessions. They will also release the dataset that they curated for the aforementioned instruction tuning.

**Strengths:**

1. Extensive evaluation: authors evaluate their multimodal tool-based reasoning approach with a large number of tools on existing as well as their own benchmark and compare it with other SOTA LMM approaches. I think their exhaustive evaluation would be useful for the community moving forward.
2. I appreciate the authors' commitment to reproducibility and open-sourcing.
3. The paper is overall well-written and easy to follow.

**Weaknesses:**

1. Limited novelty: Despite extensive eval, the work's novelty is limited especially from the methodology standpoint since neither instruction tuning nor the use of multimodal tools for reasoning are novel methods.
2. Opensource instruction tuning dataset is one the contributions of this work. However, since the dataset itself was not human evaluated and GPT generated, it is unclear how much hallucination does it contain.
* would be good to add a limitation section in the paper and discuss this.
* would also be useful to provide dataset stats (# of instructions, # tools, # instructions/tool etc.) for completeness and perhaps human evaluation for a subset?
3. It is unclear whether addition of a new tool would require doing the instruction tuning from scratch or can the model  be fine-tuned for just those new instructions? Also unclear how the assistant deals with contradicting grounding information from two tools. For instance, if detection and segmentation tools disagree about the presence of a particular object, what does the assistant do?

These points prevent me from championing the paper.

**Questions:**

- Table 5: What does “all tools + GPT4” mean? Did the authors mean GPT4Tools?
- Table 4,5: Would have been great to see performances of atleast some non-tool based but multimodal models’ performances on Llava-bench just to see how they compare with tool-based approaches on this benchmark.
- Is it also possible to provide comparisons with GPT4-v (using visual inputs) for all benchmarks?
- Table 7: The authors should add model sizes for all models if possible.
- Instruction tuning training details are missing from the paper and the appendix.
- Would be great to see some failure cases in the paper/appendix to understand the limits of the agent's reasoning capabilities.

---

> ### Author Response · Authors · 2023-11-23
> **Responses to Reviewer #3S2M (part 1)**
>
> Thanks for your valuable comments. We first have a brief summary of your concerns and our responses, followed by detailed responses.
>
> ## Brief Summarization:
>
> 1. **Concerns about the LMM+Tools+Train designs of our framework.**
>
>     R: We show that our choice is the most reasonable way to improve LMM, i.e., LMM + Tools + Instruction Tuning, avoiding the long context window as in prompt-based methods,  enabling image context in chain ways and enabling tool result aggregation. We added a new paragraph in our revised paper to discuss the design in Sec. E.
>
> 2. **Concerns about the data filter details to reduce hallucinations.**
>
>     R: We added a new paragraph to describe the details of the data processing in Sec. A.
>
> 3. **The incremental learning performance of tool use.**
>
>     R: We added a new experiment to show the performance of LLaVA-Plus under incremental learning settings in Sec. C.
>
>
> ## Detailed Responses:
>
> - **W1: Limited novelty**: Despite extensive eval, the work's novelty is limited especially from the methodology standpoint since neither instruction tuning nor the use of multimodal tools for reasoning are novel methods.
>
> **Response:** Thank you for acknowledging the thoroughness of our evaluations. We appreciate your perspective on the novelty of our work and would like to further clarify the unique aspects of our approach. While some components of our methodology may not be novel in isolation, the way we have combined them to enhance LMMs with tool use represents a significant and innovative contribution to the field. Our approach not only enhances the performance of multimodal models but also addresses key limitations in current tool integration strategies.
>
> A more detailed discussion is available in the **Responses to Common Concerns**.
>
> - **W2.1**: Opensource instruction tuning dataset is one the contributions of this work. However, since the dataset itself was not human evaluated and GPT generated, it is unclear how much hallucination does it contain. would be good to add a limitation section in the paper and discuss this.
>
> **Response***: Thank you for highlighting the importance of data quality, especially in the context of our opensource instruction tuning dataset. We recognize the potential for hallucination in datasets generated by models like GPT, and have taken several measures to ensure the integrity and usefulness of our data.
>
>
> _Removal of Apologetic Responses_: We have filtered out responses containing phrases like “I’m sorry”, which often indicate uncertainty or errors in model-generated content.
>
> _Exclusion of Language Model Content_: Entries containing references to “AI language model” or “language AI model” have been removed to maintain focus on instructional content rather than meta-discussions about the AI itself.
>
> _Validation of CLIP-Retrieval Data_: For data pertaining to CLIP-Retrieval, we ensured that only items containing correct answers, as verified against ground-truth labels, were retained. Our selection criteria involve keeping items that match any of the listed ground truths.
>
> _Accuracy in Object Detection and Segmentation Data_: We have excluded data where the provided answers contain bounding boxes that do not align with the established ground truths, ensuring accuracy in object detection and segmentation tasks.
>
> Thanks for the suggestions. We have added a new paragraph in Sec. A of the Appendix to discuss the filter rules. Acknowledging your valuable suggestion, we have added a paragraph at the end of the main text of the revised version of our paper to discuss these limitations.
>
> - **W2.2**: would also be useful to provide dataset stats (# of instructions, # tools, # instructions/tool etc.) for completeness and perhaps human evaluation for a subset?
>
> **Response**: Thank you for your valuable suggestion regarding the provision of dataset statistics. To address this, we have included comprehensive details about our dataset in Table 2 of our paper. This table presents a thorough breakdown of the dataset, including the number of instructions and the distribution of these instructions across the different tools we utilized. Regarding the suggestion for human evaluation of a subset of the dataset, we acknowledge its importance in validating the quality and relevance of the data. While this was not included in the current scope of our study, it is certainly a valuable consideration for future work to further establish the dataset's reliability and applicability.

---

> ### Author Response · Authors · 2023-11-23
> **Responses to Reviewer #3S2M (part 2)**
>
> -  **W3.1**: It is unclear whether addition of a new tool would require doing the instruction tuning from scratch or can the model be fine-tuned for just those new instructions?
>
> **Response**: Thank you for your insightful question regarding the incorporation of new tools or skills into our model. We conducted an experiment to understand our model's incremental learning capabilities.
>
> *Impact of Fine-tuning on Specific Tools*: When fine-tuning LLaVA-Plus on a particular tool, we observed that while the model's performance in that specific area improves, it adversely affects its abilities in handling other tools. For instance, after fine-tuning solely on grounding data, the model's proficiency in tagging, captioning, and OCR significantly diminished. This is evident from the following results:
>
> | **model** | **grounding** | **tagging** | **caption** | **OCR** |
> | --- | --- | --- | --- | --- |
> | **LLaVA-Plus (4 tools)** | 0.77 | 1.00  | 0.83  | 1.00  |
> | **LLaVA-Plus (4 tools -> grounding)** | 0.93  | 0.02  | 0.36  | 0.00  |
>
> *Distinguishing Between Skills and Tools*: It's crucial to differentiate between 'skills' and 'tools' in our model's context. Skills represent the model's capabilities and may require a complete retraining process for effective integration of new skills. Conversely, the model is more adaptable when it comes to incorporating upgraded tools. For instance, replacing an existing tool with its advanced version (e.g., upgrading from CLIP-B to CLIP-L) can be done seamlessly without necessitating a full retraining of the model.
>
> Therefore, if the goal is to add a new skill, a comprehensive fine-tuning across all tools would be necessary to maintain the model's overall proficiency. On the other hand, upgrading tools can be achieved with minimal impact on the model’s existing capabilities.
>
>
> - **W3.2**: Also unclear how the assistant deals with contradicting grounding information from two tools. For instance, if detection and segmentation tools disagree about the presence of a particular object, what does the assistant do?
>
> **Response**:
> - Thank you for your insightful questions regarding our approach to question rewriting and conflict resolution in tool use.
> In our model, each skill is associated with a specific tool, effectively eliminating the possibility of tool conflicts within the same skill. However, conflicts can arise when dealing with different skills due to the inherent ambiguities in human instructions. For instance, a grounding task might yield varying outputs like a simple box or a box with an accompanying mask. To mitigate such conflicts, our data processing pipeline focuses on two main strategies:
> we try to avoid ambiguous questions that could leads to conflicts when constructing instruction data. There are two approachs we used during our data processing pipeline.
> 1. **Diversifying Questions**: We intentionally diversify the questions in our dataset to reduce ambiguities and improve tool use accuracy. This approach helps in clearly defining the task for each tool, thereby reducing the likelihood of conflicting interpretations.
> 2. **Quality Control in Data Generation**: Our data is primarily generated using GPT4V, which, while powerful, can produce hallucinations or undesirable data items. To ensure data quality, we apply heuristic methods to filter out low-quality data:
>    1. Removal of Apologetic: We have filtered out responses containing phrases like
> “I’m sorry”, which often indicate uncertainty or errors in model-generated content.
>    2. Exclusion of Language Model Content: Entries containing references to “AI language model”
> or “language AI model” have been removed since we plan to train a multimodal model
> rather than a pure language model.
>    3. Validation of CLIP-Retrieval Data: For data pertaining to CLIP-Retrieval, we ensured that
> only items containing correct answers, as verified against ground-truth labels, were retained.
> Our selection criteria involve keeping items that match any of the listed ground truths.
>    4. Accuracy in Object Detection and Segmentation Data: We have excluded data where the
> provided answers contain bounding boxes that do not align with the established ground
> truths, ensuring accuracy in object detection and segmentation tasks.
>
> By implementing these measures, we aim to minimize conflicts and ambiguities, ensuring that the model produces reliable and accurate outputs in line with the intended instructions.

---

> ### Author Response · Authors · 2023-11-23
> **Responses to Reviewer #3S2M (part 3)**
>
> - **Q1**: Table 5: What does “all tools + GPT4” mean? Did the authors mean GPT4Tools?
>
> **Response**: Thanks for the question, which help clarify important aspects of our research. In Table 5, “all tools + GPT4” refers to a setup where we integrate the outputs from all the selected tools and then feed this combined information to GPT4. The purpose of this process is for GPT4 to aggregate the diverse inputs and synthesize an answer. This approach is part of an oracle experiment designed to demonstrate the potential effectiveness of tool use in enhancing the performance of a language model like GPT4 in processing and synthesizing multimodal information.
>
> - **Q2**: Table 4,5: Would have been great to see performances of at least some non-tool based but multimodal models’ performances on Llava-bench just to see how they compare with tool-based approaches on this benchmark.
>
> **Response**: Regarding your second query, we have included performance data for the plain LLaVA model in Tables 4 and 5. The plain LLaVA, which is trained on visual instruction data without the integration of any additional tools, serves as a baseline for comparison. This model represents a non-tool based, multimodal approach. By contrasting its performance with that of our tool-enhanced models, we aim to showcase the relative effectiveness and advantages of incorporating tools in multimodal modeling. These comparisons provide valuable insights into how tool-based approaches can enhance the capabilities of multimodal models in processing complex, multimodal datasets.
>
> - **Q3**: Is it also possible to provide comparisons with GPT4-v (using visual inputs) for all benchmarks?
>
> **Response**: Thanks for your question. In response to your question, we have indeed conducted a comparative analysis of our model with GPT4-v on the LLaVA-Bench, as shown in the Table below. These results indicate that GPT4-v performs significantly better than both LLaVA and LLaVA-Plus, even though LLaVA-Plus has been augmented with additional tools.
>
> | Model | Conv | Detail | Reasoning | ALL |
> | --- | --- | --- | --- | --- |
> | LLaVA | 82.0 | 69.1 | 92.6 | 81.2 |
> | LLaVA-Plus | 81.6 | 74.5 | 95.7 | 83.9 |
> | GPT4V | 67.3  | 104.3 | 108.4 | 93.3 |
>
> Table R2.1 Results on LLaVA-Bench
>
> Regarding testing on other benchmarks like the SEED-Bench, which contains over 10,000 test examples, we acknowledge the potential value of this comparison. However, conducting such extensive tests with GPT4-v is resource-intensive. We will consider testing on GPT4-v for these benchmarks if the API becomes more accessible and cost-effective.
>
> - **Q4**: Table 7: The authors should add model sizes for all models if possible.
>
> **Response**: Thank you for your valuable suggestion regarding the inclusion of model sizes in our analysis. We have updated Table 7 in our paper. A separate column has now been added to display the size of each model, to the best of our knowledge and available information.
>
> - **Q5**: Instruction tuning training details are missing from the paper and the appendix.
>
> **Response**: Thank you for pointing out the need for more detailed information regarding instruction tuning training in our study. To address this, we have introduced a dedicated section in our paper, specifically in Section 2.1, where we comprehensively describe the visual instruction tuning process utilized in the LLaVA model. In this section, we elaborate on the format and structure of the LLaVA data, which is a crucial component of our instruction tuning methodology.
>
> - **Q6**: Would be great to see some failure cases in the paper/appendix to understand the limits of the agent's reasoning capabilities.
>
> **Response**: Thanks for the valuable question. We have added some failure cases in Figure 16 and Figure 17 to show the limits of the agent’s reasoning capabilities. In Figure 16, the model demonstrates a limitation in comprehending the user’s intent. The task involved detecting an object likely to contain a cold beverage, yet the model defaulted to a generic caption “correspond object” for Grounding DINO, failing to specify the object in question. This particular example is adapted from DetGPT.
> In the case depicted in Figure 17, the model struggles with understanding the spatial relationship between two detected objects. Intriguingly, the model's performance in this regard is inconsistent; at times, it successfully discerns and responds accurately to the spatial query, but it also exhibits instances of failure. This variability underscores a need for further refinement in the model's spatial reasoning capabilities. Thanks again for your suggestions, which make our paper more comprehensive.

---

### Official Review · Reviewer_FDvd · 2023-11-01

**Soundness:** 2 fair
**Presentation:** 3 good
**Contribution:** 2 fair
**Rating:** 3
**Confidence:** 5

**Summary:**

This work focuses on the development of LLaVA-Plus, an approach that enhances the capabilities and flexibility of LMMs. LLaVA-Plus enables LMMs to leverage a wide range of skills from a skill repository, allowing them to perform various visual tasks. The approach facilitates skill-oriented dialogues, where the LMM initiates requests to call appropriate tools from the skill repository and aggregates the tool execution results. This approach expands the capabilities of LMMs and improves their engagement in human-AI interactions.

**Strengths:**

The significant contributions mostly lie in the data perspective, while the training algorithm and the model architecture are basically following the previous work. The authors create a new multimodal instruction-following tool using data, integrating lots of real-world tools (skills), like detectors, OCR, image generators, et al. The created dataset is useful to train multimodal language models to possess the ability to use pre-selected tools and perform better on downstream tasks.

**Weaknesses:**

1.  Lack of novelty. The paper feels more like an industry paper, which has heavy data engineering work to improve performance, instead of a research paper that has novel insights and approaches compared to previous work. Basically, all the design choices in this paper can be anticipated and do not provide too many insights.

2. Lack of flexibility. If I understand correctly, once new tools/skills are added, the models must be retrained on the augmented dataset to master this new tool. Is this correct? If this is the case, the overall approach can only teach the model to learn to use a fixed set of tools, instead of "learning to use tools" and "becoming a good agent that can effectively invoke external tools provided in context".

3. The tool selection doesn't convince me. I do not understand why a vision-language model, which can directly perceive the visual world, needs to be augmented with some "visual tools" and "vision-language tools", like OCR, Captioning model.

4.  Lack of ablation study and in-depth analysis. This paper doesn't show concrete evidence for the benefits of tool use. Only the absolute performance is reported. I would suggest reporting the tool usage rate and also conducting an ablation study to investigate what is the benefit coming from, the tool using ability, or just adding more tool data during the instruction fine-tuning stage.

**Questions:**

Please see the Weaknesses section.

---

> ### Author Response · Authors · 2023-11-23
> **Responses to Reviewer #FDvd (part 1)**
>
> Thanks for your valuable suggestions. We first have a brief summary of your concerns and our responses, followed by detailed responses.
>
> ## Brief Summarization:
>
> 1. **Concerns about the LMM+Tools+Train designs of our framework.**
>
>     Response: We show that our choice is the most reasonable way to improve LMM, i.e., LMM + Tools + Instruction Tuning, avoiding the long context window as in prompt-based methods,  enabling image context in chain ways and enabling tool result aggregation. We added a new paragraph in our revised paper to discuss the design in Sec. E.
>
> 2. **Concerns about the tool rate.**
>
>     Response: We added new experiments to show the effectiveness of our method in Sec. C.
>
> ## Detailed Responses:
>
> 1. **W1. Lack of novelty.** The paper feels more like an industry paper, which has heavy data engineering work to improve performance, instead of a research paper that has novel insights and approaches compared to previous work. Basically, all the design choices in this paper can be anticipated and do not provide too many insights.
>
>
> **Response:** Thank you for your insightful feedback regarding our work on integrating Large Multimodal Models (LMM) with tools. We believe that the novelty of our work lies not only in the conceptualization of integrating LMM with tools, but also in the effective execution and demonstrable results that significantly enhance LMM capabilities.
>
> We are the first to explore the combination of LMM, tools, and training (LMM+tools+train), a path distinct from the widely used LLM+tools+chain approach. This novel integration demonstrates its effectiveness. Additionally, our proposed visual instruction tuning approach and data format is an innovative contribution that enhances the utility and clarity of LMM outputs.
>
> A more detailed discussion is available in the **Responses to Common Concerns**. Thanks again for your valuable comments.
>
> 2. **W2: Lack of flexibility.** If I understand correctly, once new tools/skills are added, the models must be retrained on the augmented dataset to master this new tool. Is this correct? If this is the case, the overall approach can only teach the model to learn to use a fixed set of tools, instead of "learning to use tools" and "becoming a good agent that can effectively invoke external tools provided in context".
>
> **Response:** Thank you for your valuable feedback regarding the flexibility of our model in learning and utilizing new tools and skills.
>
> *Clarifying Skills vs. Tools*: It's important to distinguish between 'skills' and 'tools' within our model's framework. Skills are the model's capabilities, which can be achieved using various tools. When a new skill is required, it is true that the model needs to be retrained to effectively integrate and master this new skill. However, when it comes to replacing or upgrading tools, LLaVA-Plus exhibits greater flexibility. For example, if a more advanced version of a tool (like upgrading from CLIP-B to CLIP-L or switching from EasyOCR to Google OCR API) becomes available, we can seamlessly integrate this new tool into the model without the need for retraining. This approach ensures continuous improvement in the model's performance with minimal retraining requirements.
>
> *Flexibility vs. Chain Methods*: We acknowledge that our approach might not be as inherently flexible as chain methods in terms of tool use. However, the chain method comes with its own set of challenges, such as tool conflicts during inference. LLaVA-Plus is designed to avoid such conflicts, thereby ensuring more stable and reliable performance.
>
> *Open-Source LMM Initiative*: LLaVA-Plus is developed as an open-source LMM to further easily enhance the model capabilities. Tool chaining methods frequently require extensive context to describe tools, reducing overall model performance, especially when dealing with a large number of tools. Our framework is designed to bypass these issues, providing a more efficient and effective solution for integrating and utilizing a large tool set.
>
> We appreciate your insights and hope this clarification highlights the strengths and future potential of our approach.

---

> ### Author Response · Authors · 2023-11-23
> **Responses to Reviewer #FDvd (part 2)**
>
> 3. **W3: The tool selection doesn't convince me.** I do not understand why a vision-language model, which can directly perceive the visual world, needs to be augmented with some "visual tools" and "vision-language tools", like OCR, Captioning model.
>
> **Response:** Thank you for raising an important question about our choice of tools in augmenting a vision-language model. We understand your concern regarding why a model capable of directly perceiving visual data would require additional "visual" and "vision-language" tools.
>
> There are primarily two strategies to improve a model's capabilities: one is by enriching the training dataset, and the other is by integrating specialized external tools for specific tasks. While the former can be effective, it is often resource-intensive and costly. Our focus has been on exploring the latter approach – utilizing expert tools – to strengthen the capabilities of LMM in a more efficient manner.
>
> We select five tools to *cover the essential image understanding abilities*: RAM for image classification, Grounding DINO for object detection, Grounded-SAM for instance segmentation, BLIP2 for image caption, and EasyOCR for OCR. These tools are specifically designed for their respective tasks and may offer image representations that are complementary to LLaVA.
>
> By integrating these expert tools, we can significantly enhance the model’s performance in each of these key areas of image understanding. In our experiments, as shown in Table 5, we demonstrate that the incorporation of these tools not only improves the model's performance in tasks like tagging and captioning but also adds new dimensions to its image understanding capabilities. The selection of these tools was driven by the objective to cover a wide range of essential image understanding abilities. Their integration into our vision-language model is not a redundancy, but rather a strategic enhancement to leverage the strengths of specialized tools, thereby enabling the model to perform a variety of tasks with greater accuracy and efficiency.
>
> 4. **W4: Lack of ablation study and in-depth analysis.** This paper doesn't show concrete evidence for the benefits of tool use. Only the absolute performance is reported. I would suggest reporting the tool usage rate and also conducting an ablation study to investigate what is the benefit coming from, the tool using ability, or just adding more tool data during the instruction fine-tuning stage.
>
> **Response**: Thank you for your constructive feedback on our paper. In response to your suggestions, we have expanded our analysis to include more concrete evidence demonstrating the benefits of tool integration in our model.
>
> *Reporting Tool Usage Rate*: To address your point about reporting the tool usage rate, we have added new experimental data in Table 13 of Section C in the Appendix of our revised paper. This table provides a detailed breakdown of the tool usage rates, offering a clearer insight into how each tool contributes to the overall performance of our model.
>
> *Examples of Tool Integration Benefits*: We have also included several examples in our paper to illustrate the advantages of integrating tools into the LLaVA-Plus model. A notable example is depicted in Figure 6, where LLaVA-Plus demonstrates the capability to detect and segment objects, a functionality that the original LLaVA model lacks.
>
> *Expanding on New Scenarios*: In addition to these examples, we have dedicated Section D of our paper to present a variety of new scenarios where the tool-enhanced model excels. These scenarios include diverse applications such as meme understanding, image generation, image editing, interactive segmentation, and analyzing social media posts. These examples serve to showcase the versatility and improved performance of our model in handling complex, real-world tasks. Through these additional experiments and expanded discussions, we aim to provide a comprehensive understanding of how tool integration not only enhances the model's existing capabilities but also enables it to tackle new and diverse challenges effectively.

---

### Author Response · Authors · 2023-11-23
**Paper Revision**

Thanks for all the reviewers' valuable comments. We have updated a revision of our paper. The updated paragraphs are marked in purple (violet). Below is a summary of the key changes:

- **New Section on LMM vs. LLM and Training vs. Chaining (Sec. E)**: Added to discuss the advantages of LMM+Tools over LLM+Tools, and training methods over chaining for tools, in response to reviewers' suggestions.
- **New Experiments on Tool Use Rate (Sec. C)**: Incorporated to provide more in-depth analysis, following recommendations from reviewers.
- **Data Filter Rules Discussion (Sec. A)**: Introduced to elaborate on our data filtering methodology, as suggested by reviewers 3S2M and aMGS.
- **Paragraph on Paper Limitations**: Added at the end of the main text to address potential shortcomings, acknowledging reviewer 3S2M's advice.
- **Incremental Learning Experiment (Sec. C)**: Included to explore the model's adaptability, in line with reviewer 3S2M's guidance.
Model Size in Table 7: Updated to reflect the model size, responding to reviewer 3S2M's input.
- **New Section on LLaVA (Sec. 2.1)**: Created to provide a comprehensive background on Visual Instruction Tuning in LLaVA, as per suggestions from reviewers 3S2M and xZda.
- **Failure Cases in Reasoning (Tables 16 and 17)**: Presented to highlight the model's limitations in reasoning tasks, based on reviewer 3S2M's proposal.
- **Clarifications in Footnotes**: Added to elucidate the terms "planning" and "tools" in our paper, following reviewer aMGS's suggestion.
Revised Sentences for Clarity: Adjusted throughout the main text to enhance readability and understanding, thanks to reviewer aMGS's constructive feedback.
- **Expanded Figure 2**: Modified for better clarity and reader comprehension, in line with the feedback from reviewers xZda and aMGS.
- **Revised Figure 3 for Undersea Example**: Altered to more accurately represent the intended scenario, as suggested by reviewer aMGS.
- **New Section Comparing Models (LLaVA and ToolFormer)**: Added to provide a comparative analysis, following reviewer xZda's recommendation.

---

### Author Response · Authors · 2023-11-23
**Responses to Common Concerns**

Thanks for the valuable comments from all reviewers. We would like to first address the common concerns.


1. **Novelty and Impact of LMM Integration with Tools:**

    While instruction tuning and tool use in isolation are not novel concepts, our work represents the first attempt to specifically tune a Large Multimodal Model (LMM) for tool use. This approach goes beyond traditional methods by effectively integrating a variety of tools with an LMM, thereby expanding the abilities of these tools and enhancing the model’s overall capabilities. This unique integration is demonstrated through extensive experiments on several public benchmarks.


2. **Motivation and advantages of our choices:**

    We would like make some comparisons of some different solutions:

    **a) LMM+Tools vs. LLM+Tools**: Our LMM receives image features as inputs, enabling a more comprehensive understanding of images compared to traditional Language Large Models (LLMs) with tools. This is evident in Table 4(a), where our model outperforms GPT4Tools (an LLM with tools). Additionally, the LMM's ability to visually perceive tools allows it to correct and refine tool outputs more effectively, as illustrated in Tables 11 and 12.

    **b) Training for Tools vs. Chaining for Tools**: Unlike the chaining method, which often leads to tool conflict during inference, our model is trained to use the appropriate tools in most scenarios. The chaining approach typically requires extensive context windows for instructions, making it unsuitable for many open-source models. Our training method overcomes these limitations, offering a more efficient and practical solution.

    **c) Context Cost in Chining:** In Section E, we provide an example of the popular Visual ChatGPT. For a single query, it requires over 2500 tokens, a count that exceeds the processing capacity of most open-source language models like LLaMA. This case study underscores the practical challenges and limitations faced by current models in handling extensive tool integrations and further highlights the significance and applicability of our approach.

    In summary, while some components of our methodology may not be novel in isolation, the way we have combined them to enhance LMMs with tool use represents a significant and innovative contribution to the field. Our approach not only enhances the performance of multimodal models but also addresses key limitations in current tool integration strategies.

---

### Meta-Review · Area_Chair_3HWw · 2023-12-07

**Metareview:**

The authors introduce LLaVA-Plus, based on a variety of large vision and vision-language pre-trained models. LLaVA-Plus undergoes training with multimodal instruction-following data, applicable for visual understanding, visual generation, external knowledge retrieval, and their combinations. Additionally, the image query is grounded in and remains actively involved throughout the human-AI interaction sessions.

Reviewers FDvd, 3S2M, and aMGS have expressed concerns about the novelty of the work. Having thoroughly reviewed the rebuttal and the feedback from the reviewers, I find that these concerns remain unresolved. Therefore, I recommend a reject rating

**Justification For Why Not Higher Score:**

As mentioned above, reviewers FDvd, 3S2M, and aMGS have raised concerns about the novelty of the work. Having thoroughly reviewed the rebuttal and the feedback from the reviewers, I find that these concerns remain unresolved.

**Justification For Why Not Lower Score:**

N/A

---

### Decision · Program_Chairs · 2024-01-16

Reject